# Large-scale emergence of regional changes in year-to-year temperature variability by the end of the 21st century

Dirk Olonscheck [1,2✉], Andrew P. Schurer[1], Lucie Lücke[1] & Gabriele C. Hegerl [1]

Global warming is expected to not only impact mean temperatures but also temperature variability, substantially altering climate extremes. Here we show that human-caused changes in internal year-to-year temperature variability are expected to emerge from the unforced range by the end of the 21st century across climate model initial-condition large ensembles forced with a strong global warming scenario. Different simulated changes in globally averaged regional temperature variability between models can be explained by a trade-off between strong increases in variability on tropical land and substantial decreases in high latitudes, both shown by most models. This latitudinal pattern of temperature variability change is consistent with loss of sea ice in high latitudes and changes in vegetation cover in the tropics. Instrumental records are broadly in line with this emerging pattern, but have data gaps in key regions. Paleoclimate proxy reconstructions support the simulated magnitude and distribution of temperature variability. Our findings strengthen the need for urgent mitigation to avoid unprecedented changes in temperature variability.

[1] School of GeoSciences, University of Edinburgh, Edinburgh EH9 3JW, UK. [2] Max Planck Institute for Meteorology, 20146 Hamburg, Germany.
✉email: dirk.olonscheck@ed.ac.uk

The past and projected trajectory of global warming are relatively well understood, but changes in climate variability and associated climate extremes remain uncertain. Changes in temperature variability are at least as important as the change in mean temperature because increased variability poses a greater risk to species and human society than global warming[1,2]. Despite this relevance, we know little about changes in temperature variability[3] due to the difficulty of reliably quantifying changes in variability based on instrumental records, single climate model simulations or multi-model ensembles from Climate Model Intercomparison Projects (CMIPs). The challenge of disentangling the forced response and changes in internal variability with these traditional tools results in inconclusive estimates of projected change, ranging from no change[4], slight global-mean decreases[5–7], to regional increases in temperature variability[7–9]. We use the unprecedented opportunity to investigate changes in temperature variability with single-model initial-condition large ensembles (SMILEs[10]) from multiple models and contextualise these recent and future results with estimates from instrumental records, paleoclimate proxies, and model simulations of the past climate such as the Last Millennium Ensemble of the Community Earth System Model (CESM1-CAM5 LME[11]). SMILEs allow us to separate internal variability from the variability due to external forcing[10,12], making it possible to derive continuous and robust estimates of internal temperature variability.

We here show that anthropogenically forced changes in internal interannual temperature variability are projected to emerge from the unforced range of internal variability over the 21st century. While future globally averaged regional temperature variability decreases only slightly across the model average, the contrasting pattern of increasing variability over tropical land and decreasing variability in high latitudes is projected to become much more pronounced under strong global warming.

## Results

**Paleoclimate and instrumentally recorded evidence.** To quantify the evolution of regional near-surface air temperature variability from 850 to 2100 CE, we compare estimates of internal temperature variability from instrumental records, proxy reconstructions and several model simulations (see the "Methods" section, Fig. 1). To do so, we interpolate all data and simulations to a nominal spatial resolution of 1° × 1°. After removal of the forced response represented by the SMILE means, the global average of local standard deviations from observations and model simulations agree in magnitude with a central value of about 0.44 and 0.47 °C, respectively, which is relatively stable since 850 CE (Fig. 1a, Supplementary Fig. 1 for the standard deviation of global mean temperature). These are lower-bound estimates of the true global value because primarily highly variable high-latitude regions are masked out to account for data gaps and uncertain data in HadCRUT5 (see the "Methods" section). To evaluate the magnitude of variability in global mean temperature for the last millennium, we compare the natural variability derived from the multiproxy database PAGES2k with the CESM1-CAM5 LME (Fig. 2a), the only last millennium ensemble with more than 10 ensemble members that currently exists. We find that CESM1-CAM5 LME is at the upper end of the large spread from different reconstruction methods of PAGES2k, in line with findings that PAGES2k is insensitive to high-frequency variability[13,14]. In periods with strong volcanic eruptions, the simulated variability in global mean temperature substantially exceeds the variability from PAGES2k. This is explained by the smaller impact of strong volcanic eruptions such as the Samalas eruption in year 1257 in proxy reconstructions compared to the model simulation[13,15,16]. The single-forcing simulations of CESM1-CAM5 LME support

the primary importance of volcanic eruptions in driving globally averaged regional temperature variability (Fig. 2b). Volcanic forcing dominates the natural forcings over the last millennium, while orbital variations, changes in the solar cycle, natural variations in greenhouse gas concentrations and land-use/land cover change are less relevant for interannual temperature variability[14,17].

To evaluate the spatial distribution of temperature variability for the last millennium, we focus on the pattern of Northern Hemisphere summer temperature variability estimated from tree rings as compiled in the N-TREND database[18]. We find that N-TREND resembles the average variability pattern from ten models of the Paleoclimate Modelling Intercomparison Project 3 (PMIP3[19]) and CESM1-CAM5 LME (Fig. 2c–e), except for known biases of N-TREND in the Quebec region[18]. The models generally show larger variability than N-TREND especially over land, but the differences between simulated and observed estimates have a similar magnitude to inter-model differences (Supplementary Fig. 2). Overall, the broad consistency between simulated estimates across different periods of the past and the estimates from paleo-proxies and instrumental records increases confidence in the ability of global climate models to simulate past and future interannual temperature variability.

Estimating changes in temperature variability from instrumental records is challenging because observations are a single, non-stationary realisation within the range of possible trajectories[10]. Here we use the similarity between the instantaneous ensemble variability and the temporal standard deviation after detrending to consistently compare the SMILE simulated changes in temperature variability with observed changes[7] (see the "Methods" section, Fig. 1, Supplementary Fig. 3). To remove the forced signal, we detrend the instrumental records with the multi-model mean of the SMILE means. We evaluate the standard deviation ratio between the two periods 1970–2019 and 1920–1969 averaged across SMILEs and observational products (Fig. 1b, c). This result is robust to detrending the instrumental records with each individual SMILE mean instead of the multi-model mean, suggesting that the ratio of observed variability between both periods is insensitive to the model used to detrend the observations (Supplementary Fig. 4). We further find that the observed pattern of temperature variability change is consistent across datasets (Supplementary Figs. 5 and 6) and confirms the large-scale simulated pattern with increased variability on tropical and subtropical land and the central and eastern Pacific[20,21], and decreased variability at mid-latitudes. The observed changes are generally stronger than the simulated changes (Fig. 1b, c), despite the more conservative approach of quantifying variability used here (Supplementary Fig. 3). The observed increase in tropical Pacific temperature variability in the period 1970–2019 compared to 1920–1969 might be caused by natural multi-decadal changes in ENSO variability[22]. Differences in simulated and observed temperature variability change at high latitudes might result from uncertainty in the sparse observational data especially early in the record, or from different timing of observed sea ice loss compared to simulations[23]. Possible increases in observed temperature variability at high latitudes could be explained by temporarily increased temperature variability during the transition from ice-covered to seasonally ice-free polar oceans[7,24,25]. This is supported by many models as discussed below (compare Fig. 4a). Missing and uncertain data, especially in regions of interest such as tropical land and high-latitude oceans, limit the confidence in the observed pattern. However, the large-scale similarity between the simulated and observed pattern elsewhere increases confidence in the plausibility of projected changes in variability (Fig. 1b, c, Supplementary Figs. 5 and 6).

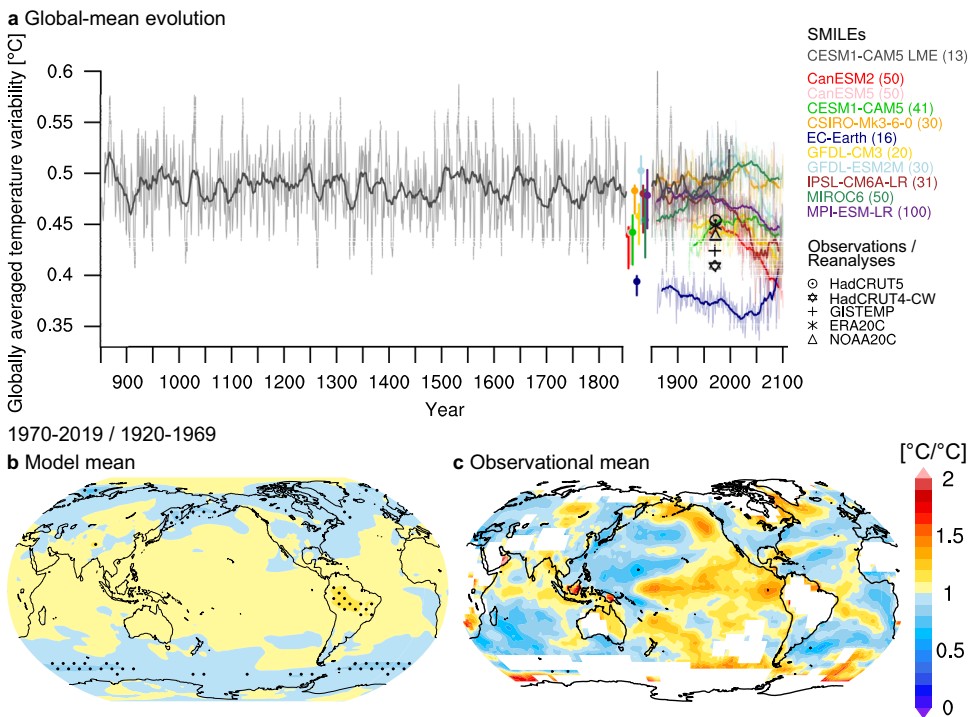

**Fig. 1 Evolution of interannual temperature variability from 850 to 2100 CE. a** Annual globally averaged regional ensemble standard deviation for each model year across SMILEs (coloured lines), including the CESM1-CAM5 Last Millennium Ensemble (grey line). Thick lines show centred 20-year running averages. The filled dots with vertical lines show the standard deviation across detrended preindustrial control simulations, and the range of preindustrial temperature variability derived from all standard deviations across consecutive overlapping 100-year running averages from each preindustrial control simulation. The climate model estimates are compared to the interannual temperature variability estimated from observations and reanalyses that were detrended with the multi-model mean of the SMILE means. All data and simulations are masked where HadCRUT5 has data gaps or insufficient data in 1920–1969 (see the "Methods" section). Compare with Supplementary Fig. 1 for the standard deviation of global mean temperature. **b** and **c** Average spatial pattern of change in temperature variability determined as the ratio between the periods 1970–2019 and 1920–1969 from **b** eight SMILEs (CanESM2 and GFDL-ESM2M only start in 1950), and **c** five observational products (ERA-20C, NOAA-20C, HadCRUT4-CW, GISTEMP, and HadCRUT5; see the "Methods" section). Variability in the observational products is calculated as standard deviation over both 50-year periods. Grid points with insufficient temporal coverage in HadCRUT5 are masked out in (**c**). Stippling marks significant changes in temperature variability at a 5% level based on an $F$-test.

**Inconsistent global mean change.** Despite some confidence in the simulated temperature variability, we find quite different model responses in globally averaged regional temperature variability change (Fig. 3a–c). Anthropogenic forcing caused by human carbon emissions has only marginally changed the mean SMILE response in globally averaged regional temperature variability in 2010–2019 compared to preindustrial, which is consistent with previous findings[5,7]. However, the SMILEs disagree on the direction of globally averaged change over the 20th century and more so, the 21st century (Figs. 1a, 3b, c, Supplementary Fig. 1). Whereas most models project a continuous decrease in globally averaged regional temperature variability, a few models project a different evolution, such as no change in variability (CSIRO-Mk3-6-0), first increasing and then decreasing variability (CESM1-CAM5, MIROC6), or first decreasing and then increasing variability (EC-EARTH, see Figs. 1a, 3g). The projected change in globally averaged regional temperature variability of most SMILEs emerges outside the range of unforced temperature variability derived from each preindustrial control simulation, but in different directions: averaged across 2080–2099, five models emerge below the range of unforced temperature variability (CanESM2, CanESM5, GFDL-CM3, GFDL-ESM2M, IPSL-CM6A-LR), while two models show an increased variability above the range of unforced temperature variability (EC-EARTH, MIROC6). Three models do not emerge at the end of the 21st century (CESM1-CAM5, CSIRO-Mk3-6-0 and MPI-ESM-LR).

**Consistent patterns of change.** Because humans do not feel changes in globally averaged temperature variability, but local changes, we investigate the patterns of regional variability change. In contrast to the inconsistent globally averaged change, we find large-scale regions where all SMILEs have a consistent sign of temperature variability change, namely on tropical land (30°N–30°S) and at high latitudes (90–50°N and 50–90°S, Fig. 3). In order to place the historical and future change in temperature variability in context, we compare the patterns of change to the magnitude of variability caused by natural external forcings. To do this, we use the CESM1-CAM5 LME. We find that the naturally forced change in 10-year averaged internal temperature variability on tropical land and at high latitudes ranges from −7.9% to 3.8% and −9.5% to 2.6% of the preindustrial control variability, respectively (Fig. 3a, see the "Methods" section). This reflects strong impacts of short-term natural external forcings such as volcanic eruptions on temperature variability, and the—on average—uniform pattern of internal temperature variability change caused by natural external forcings (Fig. 3d). However, and unlike natural external forcings, anthropogenic forcing causes a distinct pattern of temperature variability change with strong increases in temperature variability on tropical land on average +6.5% (range: 0.8–14.8%), and substantial decreases in temperature variability at high latitudes on average −6.4% (range: −18.3% to 0.4%, Fig. 3b, e), averaged across 2010–2019. According to the mean response of all SMILEs, the anthropogenically forced changes of tropical land temperature

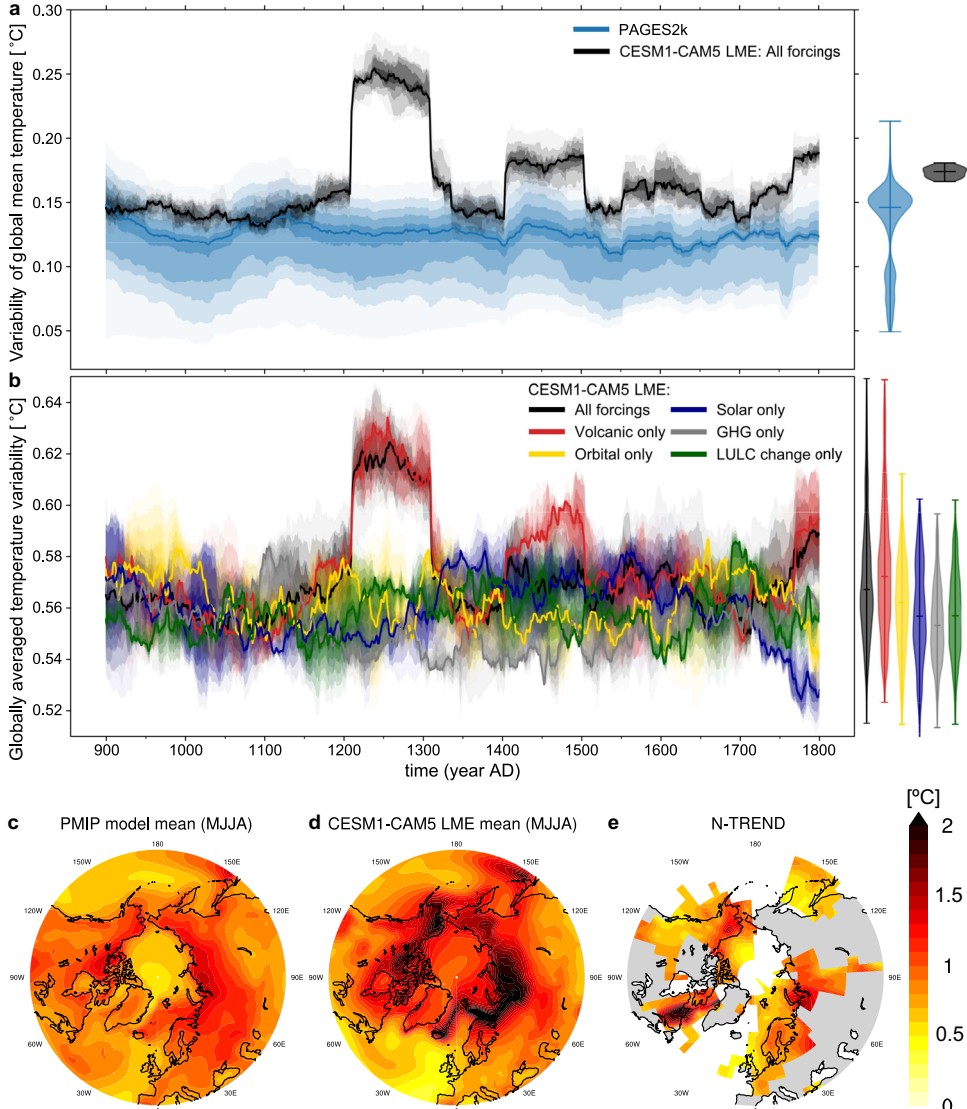

**Fig. 2 Evaluation of (naturally forced combined with internal) last millennium temperature variability in climate models. a** Variability in global mean temperature in PAGES2k compared to CESM1-CAM5 LME. The estimates are derived from the temporal standard deviation of detrended consecutively overlapping and centred 100-year periods. Note increases in variability in CESM1-CAM5 LME around large volcanic eruptions which are larger in the model than data. The shading indicates the reconstruction uncertainty for PAGES2k, and the different implementations of internal variability for the 13 ensemble members of CESM1-CAM5 LME (5th–95th percentile: lightest shading, 40th–60th: darkest shading, median: thick line). The violin plots include the complete ensemble of 100-year periods (horizontal line: median, lower and upper end: minimum and maximum). **b** Contribution of external forcings to the last millennium globally averaged regional temperature variability derived from the temporal standard deviation of consecutively overlapping and centred 100-year periods of the all-forcing and single-forcing simulations of CESM1-CAM5 LME. Shading and violin plots as in (**a**). **c–e** Spatial pattern of Northern Hemisphere last millennium temperature variability in summer (MJJA) from **c** the mean of ten PMIP simulations, **d** the ensemble mean of CESM1-CAM5 LME and **e** the paleoclimate estimate N-TREND. Grey areas in **e** indicate land with no data.

variability should already exceed the naturally forced changes in internal temperature variability at present day, which would make human influences the main driver of historic temperature variability change. The distinct pattern of regionally contrasting forced change is expected to strengthen under strong forcing scenarios SSP5-8.5[26] and RCP8.5[27] with +12.7% (range: 1.5–26.0%) increase in temperature variability on tropical land and −23.7% (range: −39.7% to −9.8%) decrease in high latitudes in 2090–2099, with a globally averaged decrease of −3.7% (range: −9.1% to 4.3%, Fig. 3c, f). The transient evolution of internal temperature variability change supports the strong multi-model agreement on increases in temperature variability on tropical land and decreases in temperature variability at high latitudes (Fig. 3g–i).

Although the SMILEs differ in the sign and magnitude of the pattern of change in small-scale regional temperature variability, the projected changes averaged across 2090–2099 confirm the common pattern of substantially decreasing temperature variability at high latitudes and increasing variability at low latitudes, especially over land (Supplementary Fig. 7). This suggests that the latitudinally contrasting pattern that is simulated for the present decade 2010–2019 (Fig. 3b, e) will become much more pronounced under strong global warming until the end of the 21st century (Fig. 3c, f), in line with previous studies[28,29]. The SMILEs all agree on substantial decreases in temperature variability in high latitudes, with a similar magnitude over the Arctic and Antarctic ocean regions and on adjacent land (Supplementary Fig. 7). This is not yet evident in observations

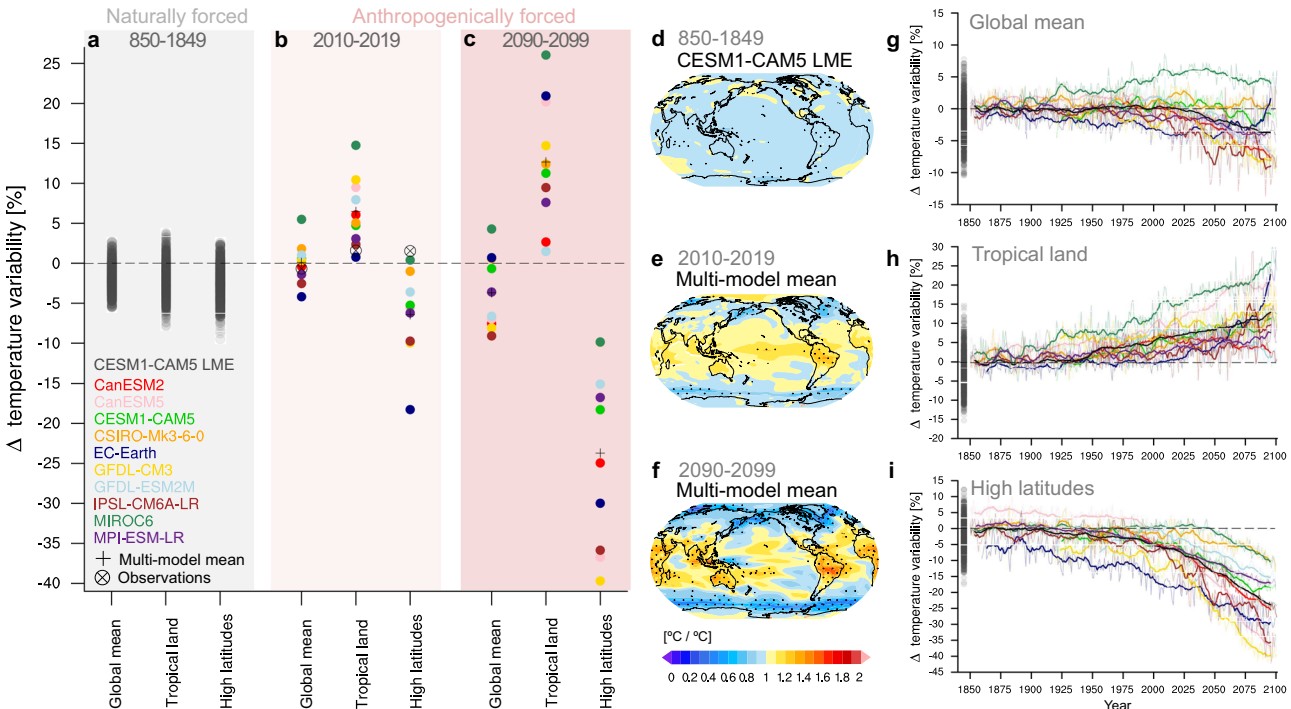

**Fig. 3 Change in internal temperature variability due to natural and anthropogenic forcing. a** Range of naturally forced globally averaged, tropical land-only (30°N–30°S) and high latitude-only (90–50°N and 50–90°S) 10-year averaged temperature variability derived from CESM1-CAM5 LME. **b** and **c** Anthropogenically forced change in globally averaged, tropical land-only and high latitude-only temperature variability averaged across **b** the present period 2010–2019, and **c** the future period 2090–2099 forced with the high emission scenarios SSP5-8.5 or RCP8.5 relative to the standard deviation of the preindustrial control simulation. The change projected by each SMILE is marked with a coloured filled dot, compared to the change in the multi-model mean and observations. The change in observed variability is calculated as the ratio between the periods 1970–2019 and 1920–1969. **d–f** Pattern of temperature-variability change averaged across **d** 1000 years of CESM1-CAM5 LME, **e** the years 2010–2019 and 10 SMILEs, and **f** the years 2090–2099 and 10 SMILEs. Stippling marks significant changes in temperature variability at a 5% level based on an *F*-test. **g–i** Transient evolution of interannual temperature variability shown as percentage change relative to the respective preindustrial control variability. Range in temperature variability derived from the CESM1-CAM5 LME (grey filled dots) compared to the past and future evolution of temperature variability derived from 10 SMILEs forced with the high emission scenarios SSP5-8.5 or RCP8.5 for **g** the global mean, **h** tropical land only, and **i** high latitudes only. Thick lines show centred 10-year running averages. The multi-model mean is shown as black line.

(Fig. 1c), in accordance with the mechanisms of temperature variability change of first increasing variability with a more seasonal ice cover in higher latitudes, accompanied by decreasing variability in regions with open ocean all year long that encompasses all the polar regions when the sea ice is gone[8] (compare Fig. 4a and Mechanisms of variability change). Furthermore, observational coverage in high latitudes is very limited. Hotspot regions of substantially increased temperature variability with multi-model agreement are the Amazon, Southeast Asia, Australia, and West Africa. The model agreement over tropical and subtropical oceans is low, which is reflected by small and insignificant multi-model mean changes. The different model responses in the tropical Pacific may be caused by strong multi-decadal changes in ENSO variability[22] and support the uncertain ENSO response to global warming caused by the interplay of many amplifying and dampening feedbacks[30,31]. We find that especially models with low globally averaged preindustrial variability (CESM1-CAM5, EC-EARTH, MIROC6, see filled dots in Fig. 1a) show substantial increases in temperature variability over low-latitude oceans under strong global warming.

**Emergence from unforced variability**. In line with the latitudinally contrasting pattern of variability change, we find robust signals of emergence of the anthropogenically forced change in interannual temperature variability from the unforced range of variability. This is evident from the fact that most SMILEs agree

on broad features of emergence at the end of the 21st century (Fig. 4b). The unforced range of variability is derived from the range of consecutive overlapping 100-year periods of the respective preindustrial control simulation of a SMILE to which the variability at the end of the 21st century is compared (see the "Methods" section). While the future temperature variability in many high-latitude regions emerges below the unforced range of variability, many low-latitude regions show an increased variability that emerges above the unforced range of variability[32], with the most consistent emergence over land areas compared to uncertain ocean patterns. In summary, despite their different evolution of globally averaged regional temperature variability, all SMILEs agree on a distinct pattern of anthropogenically forced temperature variability change with unprecedented decreases in temperature variability at high latitudes and unprecedented increases in temperature variability on tropical land.

**Mechanisms of variability change**. To better understand the pronounced changes in temperature variability at the end of the 21st century, we investigate the evolution of latitudinal temperature variability from 1850 to 2100 compared to preindustrial variability (Fig. 4a). We find that the SMILEs show a different onset of changes in variability, with most models showing substantial increase in low-latitude variability and a decrease in mid-to-high latitude variability from 2000 onward. MIROC6 shows an earlier and EC-EARTH a later onset of increase in low-latitude

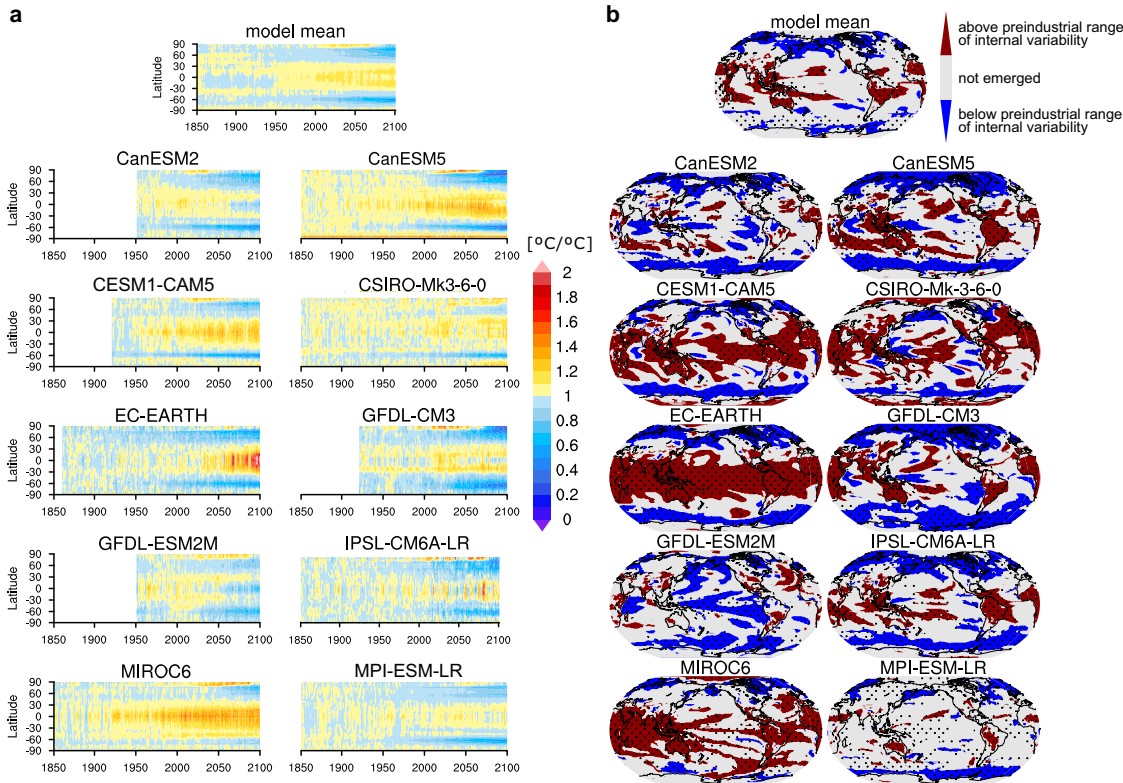

**Fig. 4 Emergence of the anthropogenically forced change in interannual temperature variability. a** Evolution of the temperature-variability change of 10 SMILEs from 1850 to 2100 versus latitude with respect to the preindustrial variability. **b** Regions of emergence with a forced increase (decrease) in variability outside the preindustrial range of variability during the period 2080–2099 are shown in red (blue), while regions that have not emerged on the 20-year average from the preindustrial range of internal variability are shown in grey. Stippling marks significant changes at a 5% level based on an *F*-test (see also Fig. 1a).

variability than the multi-model mean response, in contrast to their high-latitude decreases. We further find that all models agree on increased temperature variability over the Arctic Ocean as long as sea ice exists in the models, followed by a rapidly decreasing temperature variability when sea ice is lost.

Based on the SMILEs, we investigate the underlying mechanisms for the latitudinally contrasting pattern of present-day and future temperature variability change. Decrease in temperature variability at high latitudes is primarily caused by the loss of sea ice, as suggested before based on CMIP5[5,9] or a single climate model[6]. It is widely acknowledged that anthropogenic $CO_2$ emissions increase Arctic air temperature more than the global mean (Arctic Amplification[33]), which in turn linearly decreases Northern Hemisphere sea ice area[34,35] (Fig. 5a). In response, the decreasing sea ice area causes a substantial decrease in high-latitude temperature variability[7] (Fig. 5b), primarily due to the larger heat capacity of the open ocean compared to the insulating sea ice cover[9,36,37]. Additional causes for decreasing high-latitude temperature variability, which also extends to mid-latitudes[38], may be the decreasing meridional temperature gradient[39] and the decreasing land–sea temperature contrast[40]. The decreasing meridional temperature gradient is caused by Arctic Amplification[41] which leads to fewer warm-air intrusions into the Arctic[25]. Although strongly impacted by internal variability, the relationship between declining sea ice area and decreasing high-latitude temperature variability is consistent across SMILEs. The initial amount of sea ice and the strength of Arctic warming determines the timing and magnitude of decreases in temperature variability, largely explaining differences across models. Observations show a comparatively high initial sea ice amount with rather low Arctic temperatures (Fig. 5a) supporting a

delayed decrease in temperature variability compared to models (Fig. 3b).

In contrast to decreasing temperature variability at high latitudes, the increased variability on low-latitude land is primarily caused by a transition to drier surface types, e.g. from tropical forests to semi-arid landscapes. We use the Bowen ratio as a measure for changes in vegetation cover, and revisit the proposed relationship between the evaporative fraction and tropical temperature variability[6,9]. The Bowen ratio is the ratio between sensible and latent heat and indicates the land–surface type, and hence vegetation cover[42,43]. We compare the distribution of changes in the Bowen ratio with the distribution of changes in temperature variability over land and find a striking similarity for all SMILEs averaged across 2090–2099 compared to 1950–1959 under the strong global-warming scenarios SSP5.8-5 and RCP8.5 (Fig. 5c, d, Supplementary Fig. 8, see the "Methods" section). An increase in the Bowen ratio indicates the transition to drier surface types, e.g. from tropical forests to semi-arid landscapes. The congruity between regions with increasing Bowen ratio and the hot-spot regions of increased temperature variability supports the plausible mechanism that drier land-surface types lead to increased temperature variability, associated with soil-moisture reductions[9]. An exception are North and East African regions, which might experience increased temperature variability caused by increased precipitation and hence increased vegetation cover[44]. There is no consensus yet on the causes for the projected increased precipitation in North and East Africa[45,46]. Elsewhere in the tropics, the transition to drier land-surface types may be caused by land-use changes such as an ongoing conversion of tropical forests to agriculture or bare land[47,48], accompanied by large-scale forest fires[49,50], shifts in

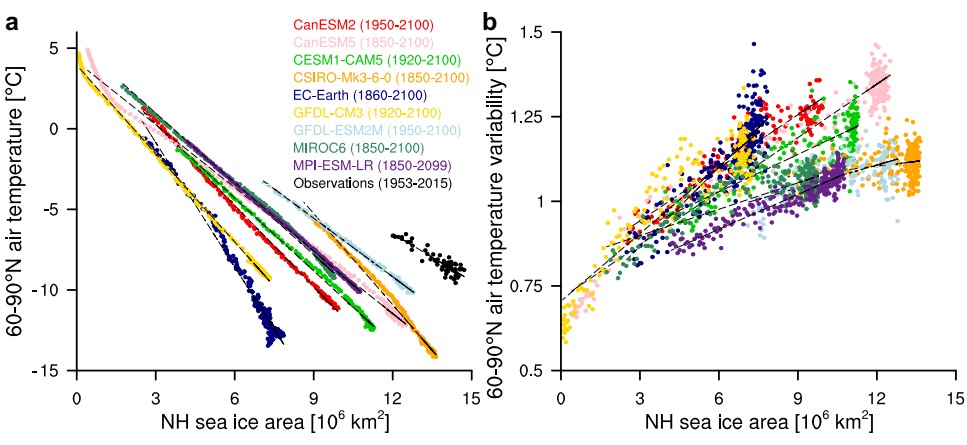

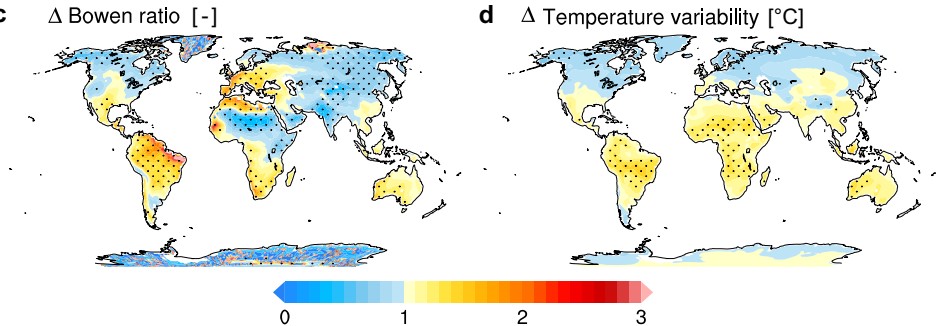

**Fig. 5 Mechanisms for the contrasting evolution of temperature variability in high latitudes and the tropics.** Relationship between annual mean Northern Hemisphere sea ice area and **a** 60–90°N mean temperature, and **b** 60–90°N temperature variability from 1850 to 2100 for the SMILEs (coloured) and, in **a** compared to observations (HadISST2 sea ice area vs. NOAA-20C, black). Each dot represents a single year and time runs from bottom right to top left in **a** and from top right to bottom left in (**b**). **c** Ratio of changes in the Bowen ratio (sensible vs. latent heat) between 2090–2099 and 1950–1959 and **d** changes in the ratio of temperature variability over land between 2090–2099 and 1950–1959 averaged across SMILEs forced with historical emissions and emission scenarios SSP5-8.5 or RCP8.5. Stippling in **c** and **d** marks significant changes at a 5% level based on an *F*-test. A high Bowen ratio indicates semi-arid landscapes and deserts, whereas a low Bowen ratio indicates tropical or temperate forests. Individual model results are shown in Supplementary Fig. 8.

precipitation patterns[6,51], and overall stronger evaporation in a warmer climate. The identified mechanisms link the unprecedented temperature-variability changes to the human interference with the climate system and explain the patterns of change that are distinct compared to past climate variability.

Quantifying changes in internal variability is one of the most data-intensive challenges in climate science, and while the much better sample of variability in SMILE data sheds much more light on these changes, it can still be questioned if the sampling is sufficient for all regions[52]. Here we use all SMILEs that are available to date to best estimate historical and future changes in temperature variability and compare them to the preindustrial and observed variability. We find that multiple sources of information from different time periods and data sources broadly support the simulated magnitude of globally averaged regional temperature variability, evidencing a stable range of past temperature variability. We further show that the projected changes in temperature variability will be larger than the comparatively stable unforced range of temperature variability over the last 1000 years, with a strong increase in temperature variability over some tropical land areas, and decreases in temperature variability at mid and high latitudes. While the decrease in temperature variability at high latitudes is driven by global warming-induced sea ice loss, the increased variability on tropical land is caused by the transition to a substantially drier land surface. The present-day onset of this distinct pattern of

forced change is supported by observations, but robust observational confirmation is not possible due to substantial data gaps and onset uncertainty in the regions of key changes. We find strong evidence from SMILEs that large-scale forced changes in internal year-to-year temperature variability will emerge from the preindustrial envelope of variability, shifting the climate into a state of unprecedented internal temperature variability. In close analogy to global warming, these accompanied unprecedented changes in temperature variability strengthen the need for urgent climate mitigation to avoid substantial human-caused increases in tropical temperature variability and related heat extremes.

## Methods

**Paleoclimate proxies, observations, and climate models.** For the last millennium, we use two paleo-proxy databases to evaluate the absolute temperature variability from model simulations for the last millennium (850–1849): the multiproxy database PAGES2k[13] and the gridded Northern Hemisphere summer temperature tree-ring dataset N-TREND[18]. For the instrumental era, we use the gridded observational datasets HadCRUT5[53] in its non-infilled mode, GISTEMP[54], HadCRUT4-CW[55,56] that incorporates HadSST4, and the reanalyses ERA-20C[57] and NOAA-20C[58] to estimate temperature variability. In comparison to paleo-proxies and observations and to quantify future changes in temperature variability, we compile a large set of model simulations: last millennium simulations (past1000) from 10 PMIP3 simulations[19], the 13-member CESM1-CAM5 Last Millennium Ensemble, and the respective ensembles of single-forcing simulations, as well as preindustrial control simulations, historical simulations, and future projections forced with the high-emission scenarios SSP5-8.5 or RCP8.5 from 10 single-model initial-condition large ensembles (SMILEs[10]). Seven SMILEs are from

the CMIP5 generation of climate models, namely CanESM2[59], CESM1-CAM5[60], CSIRO-Mk3.6.0[61], EC-EARTH[62], GFDL-CM3[63], GFDL-ESM2M[64], and MPI-ESM-LR[65]. The remaining three SMILEs are from the recent CMIP6 generation, namely CanESM5[66], IPSL-CM6A-LR[67], and MIROC6[68]. Note that for IPSL-CM6A-LR the scenario results are only based on six instead of 31 ensemble members. The last millennium simulations from PMIP3 used in this study are from the following models: bcc-csm1-1, CCSM4, CSIRO-Mk3L-1-2, FGOALS-gl, GISS-E2-R, HadCM3, HadGEM2-ES, IPSL-CM5A-LR, MIROC-ESM, MPI-ESM-P, and MRI-CGCM3. We analysed the variables near-surface air temperature, sea ice concentration, and sensible and latent heat. All data is regridded to a regular 1° × 1° horizontal grid by bilinear interpolation to allow for comparisons between observational products and different climate models. This unprecedented amount of data enables a robust quantification of past, present, and future changes in near-surface air temperature variability, and reveals the driving mechanisms of temperature-variability change.

**Detrending and standard deviation**. All last-millennium simulations and pre-industrial control simulations are linearly detrended to remove model drift. The observational products are detrended with the multi-model mean of eight SMILE means that cover the period 1920–2019. We consider this multi-model ensemble mean as the best estimate of the forced response, isolating the internal temperature variability in the observational products[69,70]. From these detrended simulations and observational products, the variability is determined by calculating the standard deviation across time:

$$\sigma(T) = \sqrt{\frac{1}{T-1}\sum_{t=1}^{T}(x_t - \bar{x})^2}, \quad (1)$$

where $\sigma$ is the standard deviation of the variable $x$, $t$ is the output interval in years, and $T$ is the length of the simulation. To derive an estimate of observed temperature variability, we calculate the temporal standard deviation across the 50-year periods 1920–1969 and 1970–2019. In contrast, for the CESM1-CAM5 LME and the 10 SMILEs, we estimate the internal variability by calculating the sample ensemble standard deviation for each year across each model's ensemble simulations, providing a continuous robust estimate of internal temperature variability through time:

$$\sigma_{\mathrm{ens}}(N,t) = \sqrt{\frac{1}{N-1}\sum_{n=1}^{N}(x_{n,t} - \overline{x_t})^2}, \quad (2)$$

where $\sigma_{\mathrm{ens}}$ is the sample ensemble standard deviation of the variable $x$, $n$ are the different ensemble simulations of a model with $N$ ensemble simulations, $t$ is the output interval in years, and $T$ is the simulation length. Following the quasi-ergodic assumption, which states that the variance of one sequence of events over time equals the ensemble variance at a given time, both approaches to estimate internal variability are consistent[7]. We average the ensemble standard deviations derived for each year over a 20-year time window in Fig. 1a, and over a 10-year time window for the periods 2010–2019 and 2090–2099 from all SMILEs as well as all overlapping 10-year chunks from CESM1-CAM5 LME in Fig. 3 to derive more robust estimates of grid-point temperature variability. We differentiate between globally averaged regional temperature variability defined as grid-point temperature standard deviation that is globally averaged (see Figs. 1a, 2b, 3a–c, g), and variability in global mean temperature defined as the standard deviation of global mean temperature values, here only used in context with PAGES2k (see Fig. 2a) and in Supplementary Fig. 1. The standard deviation of global mean temperature in Supplementary Fig. 1 shows substantially smaller variability estimates than the globally averaged regional temperature variability because globally averaging first suppresses local variability. The standard deviation of global mean temperature is unchanged or increasing (Supplementary Fig. 1), possibly caused by variability with large spatial coherence (i.e. ENSO). In contrast, the globally averaged regional temperature variability is decreasing for some models (Fig. 1a) because of the strong impact of local variability (i.e. in high latitudes).

**Emergence**. We define emergence as the globally averaged or regional deviation of the projected temperature variability for the period 2080–2099 from the full range of past unforced temperature variability. The range of unforced variability is determined by calculating the temporal standard deviation for all consecutive overlapping 100-year periods of the respective preindustrial control simulation of a SMILE. Emergence occurs if

$$\mathrm{avg}(\sigma_{\mathrm{ens}}, 2080-2099)(N,t) > \max(\sigma_{\mathrm{piC}}) \quad (3)$$

or

$$\mathrm{avg}(\sigma_{\mathrm{ens}}, 2080-2099)(N,t) < \min(\sigma_{\mathrm{piC}}) \quad (4)$$

where $N$ is the number of ensemble simulations of each model and $\sigma_{\mathrm{piC}}$ is the standard deviation of a 100-year long segment of the preindustrial control simulation. The ensemble standard deviations derived for each year are averaged for the 20-year time window 2080–2099 for robust results of the projected end-of-century temperature variability.

**Significance testing**. A significant change in temperature variability as shown by stippling is calculated using an F-test at a 5% two-sided significance level. In Fig. 4b, the F-test is partly less stringent than the emergence because the statistical significance directly depends on the length of the preindustrial control simulation of a model. For models with long preindustrial control simulations, i.e. MPI-ESM-LR, this results in larger regions with statistically significant change than regions that emerge from the larger range of sampled standard deviation.

**Observational uncertainty and masking**. To account for data gaps and uncertainty in the observational products, we mask out grid points with insufficient temporal coverage especially early in the record. We base our analysis on the non-infilled HadCRUT5 record and mask out grid points that contain less than 25 years of data in the 50-year reference period 1920–1969. This mask is also applied to the other observational products.

**Modelling of vegetation in SMILEs**. The SMILE models simulate vegetation dynamics differently[71]: GFDL-CM3, GFDL-ESM2M and MPI-ESM-LR include a fully dynamic vegetation model. CanESM2, CanESM5, CESM1-CAM5, EC-EARTH, IPSL-CM6A-LR, and MIROC6 predict vegetation phenology. CSIRO-Mk3.6.0 has prescribed vegetation properties.

## Data availability
The data used in this study are openly available in the following databases: The SMILE model output is obtained from the Multi-Model Large Ensemble Archive under accession code http://www.cesm.ucar.edu/projects/community-projects/MMLEA/. All other model output used here is accessible from the Earth System Grid Federation under accession code https://esgf-node.llnl.gov/projects/esgf-llnl/. The observations and reanalyses were downloaded under the following accession codes: GISS Surface Temperature Analysis (GISTEMP): https://psl.noaa.gov/data/gridded/data.gistemp.html; HadCRUT5 near-surface temperature data version 5.0.1.0: https://www.metoffice.gov.uk/hadobs/hadcrut5/data/current/download.html; HadCRUT4-CW: https://www-users.york.ac.uk/kdc3/papers/coverage2013//had4sst4_krig_v2_0_0.nc.gz; HadISST2.2.0.0: https://www.metoffice.gov.uk/hadobs/hadisst2/; ERA-20C reanalysis: https://www.ecmwf.int/en/forecasts/datasets/reanalysis-datasets/era-20c; and NOAA-20C version 3: https://psl.noaa.gov/data/gridded/data.20thCReanV3.html). The N-TREND dataset was downloaded under accession code https://www.ncdc.noaa.gov/paleo-search/study/19743 and PAGES2k at https://www.ncdc.noaa.gov/paleo/study/26872.

## Code availability
The code used to both process the data and create the figures for this paper can be publicly accessed at https://pure.mpg.de/pubman/faces/ViewItemFullPage.jsp?itemId=item_3332337

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

## Acknowledgements

D.O. received funding from the Alexander von Humboldt foundation and the European Union's Horizon 2020 research and innovation programme under grant agreement No 820829 (CONSTRAIN project). G.H. was funded by the NERC project 'Emergence of Climate Hazards' (NE/S004645/1) and G.H. and A.S. were funded by the NERC project GloSAT (NE/S015698/1). A.S. further received funding from a Chancellors fellowship at the University of Edinburgh. L.L. was supported by a studentship from the Natural

Environment Research Council (NERC) E3 Doctoral training partnership (NE/L002558/1). Computational resources were made available by the German Climate Computing Centre (DKRZ). We thank the US CLIVAR Working Group on Large Ensembles for providing the Multi-Model Large Ensemble Archive.

## Author contributions

D.O. designed this study, developed the methodology, and analysed the data. L.L. analysed proxy reconstructions. D.O., A.S., L.L. and G.H. interpreted the results and contributed to writing the manuscript.

## Funding

## Competing interests

The authors declare no competing interests.
