## [Peer Review File · Nature Communications]

REVIEWER COMMENTS

Reviewer #1 (Remarks to the Author):

Thank you for inviting me to review the paper: "Regional temperature variability is projected to emerge from its natural range" by Olonscheck et al.

As the authors suggest, changes in the mean state of the climate have received more attention and are better understood than changes in variability. Given that variability is an important consideration for climate extremes and adaptation, it is imperative to understand how variability might change as the planet warms. By analyzing a large set of SMILEs, observations and paleoproxies, the authors reproduce and lend additional credence to the projection that as the world warms, consistent patterns of variability change emerge across the models - variability increases over the tropical land areas and decreases over the high latitudes. The authors also briefly explore plausible mechanisms by which these changes might occur.

Recommendation: This is a well written paper. I liked reviewing it and I recommend its publication. I have a few methodological/conceptual concerns that I think would serve the authors well if addressed.

The title can perhaps be more specific. As it stands now, it does not seem to convey to the reader anything about when/why/where might variability emerge from its natural range. Might be good to have some of that information in the title.

Introduction: There is a considerable amount of literature investigating changes in interannual variability and its emergence in a warming climate. I am listing a few at the bottom that would be useful to cite for the reader.

line 23: I feel the wording is a bit awkward here. Consider rephrasing: "...which is caused by the difficulty to reliably quantify.." → "...owing to the difficulty of reliably quantifying.."

line 24: It might be useful to briefly mention why it's difficult to reliably estimate changes in variability using the tools mentioned

fig 1a label and elsewhere: The term "Globally averaged temperature variability" seems to imply that you are calculating variability at each grid point and then averaging across the globe. If the order of the operation is opposite, that is, you are averaging temperature across the globe first and then computing the variability, the term should be "variability of global averaged temperature".

figure s1a: the computation is a bit unclear to me here. Are the 100-year periods centered around each time point? For example, to calculate the estimate of variability at year 1200, are you centering a window around that year and then taking the interannual variability in that window?

figure s1b: what do the lines to the right of the plot represent?

line 61: I am a bit confused about why JJA is picked to compute variability. Since the rest of the paper deals with annual temperature, why not show the spatial distributions for annual temperature variability instead?

line 64 and figures s1c-e: I think the differences here are not negligible (e.g: the Arctic circle). I think a difference plot would show more light on the geographical patterns of differences in variability

figure s3 caption: 'extent' → 'extend'

line 158: Confidence intervals are perhaps a more refined way than range for checking whether

the future temperature variability has “emerged”. Alternatively, you could consider doing an F-test at each grid point between the 2080-2099 temperatures and the sample of preindustrial temperatures.

line 187: wording is a bit awkward here. consider rephrasing: “..caused by Arctic amplification and hence less..” → “..caused by Arctic amplification which leads to fewer..”

line 204: “confirms” is a perhaps a strong statement here, especially since you list an exception right afterwards.

Line 232: This is not a new result. What would be interesting here is a quantification of when exactly the emergence might happen. While questions about the emergence of the mean state have been widely discussed in the literature and researchers have come up with various ways of quantification of the mean, questions about the emergence of variability are harder to answer. A few studies that attempt to do this and are perhaps worth alluding to are: Deser et al. 2012, Yettella and England. 2018

Line 275: From a statistical standpoint, I don’t think the sample standard deviation is a robust estimate of internal variability. In the limit of an infinite number of ensemble members, the sample standard deviation is indeed an unbiased estimator of internal variability (i.e., it converges to the true internal variability). In reality however, we have to settle for a small number of ensemble members and this leads to large sampling uncertainty in the second and higher moments. This is why I think the yearly plots shown in Fig 1.a are so noisy and you had to smooth over a time window to get a more representative estimate of the true underlying internal variability. I would perhaps rephrase like the following: “...to get a more robust estimate of internal variability we smooth the sample estimate over xoxo time window...”. A more detailed investigation of these nuances can be found in Yettella et al. 2018

Line 280-285: As stated previously, confidence intervals/F-tests are a more formal way for checking whether the future temperature variability has “emerged”.

Line 280-285: It took me a bit of time to figure out how exactly the stats are being computed here. One thing that could help here (and in other parts of the paper where mathematical operations are involved) is to formalize with the help of equations. That way, there will be no room for ambiguity for the reader.

Figure 1a: Was there a reason why running means were calculated over a 20-year period?

citations:

Räisänen, J., 2001: CO2-induced climate change in CMIP2 experiments: Quantification of agreement and role of internal variability. *J. Climate*, 14, 2088–2104, [https://doi.org/10.1175/1520-0442\(2001\)014<2088:CICCIC>2.0.CO;2](https://doi.org/10.1175/1520-0442(2001)014<2088:CICCIC>2.0.CO;2).

Collins, M., and M. R. Allen, 2002: Assessing the relative roles of initial and boundary conditions in interannual to decadal climate predictability. *J. Climate*, 15, 3104–3109, [https://doi.org/10.1175/1520-0442\(2002\)015<3104:ATRROI>2.0.CO;2](https://doi.org/10.1175/1520-0442(2002)015<3104:ATRROI>2.0.CO;2).

Yettella, V., Weiss, J. B., Kay, J. E., & Pendergrass, A. G. (2018). An ensemble covariance framework for quantifying forced climate variability and its time of emergence. *Journal of Climate*, 31(10), 4117–4133. <https://doi.org/10.1175/JCLI-D-17-0719.1>

Deser, C., Phillips, A., & Bourdette, V. (2012). Uncertainty in climate change projections: the role of internal variability. *Climate Dynamics*, 38, 527–546. <https://doi.org/10.1007/s00382-010-0977-x>

Yettella, V., and M. R. England, 2018: The role of internal variability in twenty-first-century

projections of the seasonal cycle of Northern Hemisphere surface temperature. *J. Geophys. Res.*, 123, 13 149–13 167, <https://doi.org/10.1029/2018JD029066>.

Dwyer, J. G., Biasutti, M., & Sobel, A. H. (2012). Projected changes in the seasonal cycle of surface temperature. *Journal of Climate*, 25(18), 6359–6374.

Reviewer #2 (Remarks to the Author):

Review of Olonscheck et al. "Regional temperature variability is projected to emerge from its natural variability"

This study investigates future changes in surface temperature variability in various regions across the globe, using paleo-proxy records, observations and large ensemble model simulations. They find that by the end of the century, interannual temperature variability (defined as either the temporal standard deviation or the standard deviation across the large ensemble) decreases in high latitudes and is outside of the range seen in pre-industrial control runs, while in tropical land areas, interannual temperature variability increases. This is attributed to loss of sea-ice in high latitudes and changes in vegetation cover in the tropics (changing the split between sensible and latent heat).

I find this to be a very interesting and well-written study, with important implications. However, I have one main concern that I think needs to be discussed. The comparison with observed internal variability is achieved by removing the forced response from the multi-model 'SMILE' ensemble average. The SMILE ensemble average can be assumed to be a very good estimate of the simulated forced variability, but it is unclear how alike it is to the real-world 'true' forced response. There may be common model biases in the forced response (that exist across all models and hence are present in the multi-model mean) which when removed from the observations either leaves part of the forced trend unremoved or introduces spurious trends in the wrong direction. If the forced trend isn't fully removed, this could substantially bias the estimates of interannual variability.

Related to that point, the authors comment (L 78-79) that the temporal standard deviation is the more conservative estimate for interannual variability, supported by Figure S2. Figure S2 however shows the different estimates for internal variability where each model is detrended with its own forced response, therefore the separation between forced response and internal variability is 'correct' for that model by definition, as long as the ensemble is large enough. The same cannot be said for observations.

Unfortunately, there is no way to perfectly estimate of the real-world 'true' forced response, so I appreciate the difficulty of doing this and that choices have to be made, even if they aren't perfect. However, given the importance of estimating interannual variability for this study, I think this needs to be discussed at the very least. The impacts of this de-trending choice could be tested by checking that other methods of detrending give similar results, or perhaps checking how the results would be affected by using model X's forced response to detrend model Y (or use different models separately to detrend the observations – which will likely be worse than the MMM since the forced response of the MMM is likely to be less biased than a single model's, but I think it would good to show some examples of how different estimates of the forced response affect the results). Other than that, while some aspects of future changes in variability have been shown before (e.g. decrease in high-lat variability), the combination of paleo-proxy records, observations and large ensembles is novel and a useful contribution to our understanding of how interannual climate variability might change in the future. I would therefore support the publication of this article subject to the authors addressing my comment above. I also include a few minor comments.

Minor comments:

-L 43: where does the value 0.47 degC come from? The authors point to Figure S1a, but that figure seems to suggest values between 0.15-0.20 degC unless I misunderstand either the text or the figure. This does need to be addressed but should be easy to fix (or clarify if I have misunderstood)

-If there is space, I think the authors should consider making figure S1 part of the main manuscript as it really helps the flow of the paper and is key for understanding

-L 43-44: consider mentioning the de-trending here rather than only in the caption of the Supplementary figure

-L51-58: it isn't clear from the text whether the increased variability following volcanic eruptions in model simulations is considered realistic or not, only that PAGES2K is known to underestimate high-frequency variability, but that appears to be true throughout the period considered, not just for the volcanic part.

-L54-58 is a bit confusing without a sentence more introducing the single forcing experiments of CESM-CAM5 LME or Figure S1. I would suggest inserting "... Driving *changes* in *globally averaged natural temperature variability*..." As well as inserting "natural" in front of "greenhouse gas concentrations" L57 to emphasise that you are talking about natural variations of GHGs here (it becomes clear when reading carefully and checking the Supplementary Figure, but not all readers do that!)

-L69 and L78-79: see major comment above regarding comparison between observed temporal variability and ensemble standard deviation in SMILES

-L90: small suggestion: personally, I would say "some confidence" rather than "confidence" given the amount of available evidence, but this is up to the authors

-L 111-112: I'm a bit confused here as to what you mean by "long-term changes caused by natural variability". I would not expect solar and volcanic forcing to have truly long-term effects on variability (maybe a decade or two at most), are you talking about longer time scales than that (time scales of orbital changes) here? The next sentence suggests you look at changes over 10 years, which I would not class as long term. Please clarify.

-L 173-174: "at the expense of" is a bit confusing wording, somehow suggests a causal link between the tropical changes delaying the high latitude changes

Reviewer #3 (Remarks to the Author):

General

This paper uses an extensive set of model simulations of past and future climate to investigate the likely changes in worldwide local to regional surface air temperature variability during the late 21st century compared to the last 1000 years and the recent climate. The main conclusion of the paper is that there is evidence for, and good physical reasons for, a decrease of high latitude temperature variability and an increase of tropical land temperature variability. The physical arguments are stronger than the model evidence because not all models are fully consistent in showing increased tropical temperature variability in the later 21st century. As far as they go (see below) the statistical analyses are appropriate but probably insufficient. This topic is a suitable one for Nature Communications as it is important physically, and as mentioned in the paper, for potential impacts on tropical society. The paper needs a fair amount of

detailed, largely minor, revision. There is some over confidence in the strength of the conclusions about the global pattern of future variability based on the not totally consistent model evidence. Thus there are no conclusive statistical tests on this key topic so I strongly recommend that more work be done here. Without this, the paper may still be publishable but would not be so compelling in its results. Over confidence in the conclusions is reflected in the current wording of the Abstract. At least tentatively, I recommend publication of this paper, subject to the revisions below. My recommendation would be considerably stronger if further statistical tests confirmed the statistical significance of the pattern of modelled late 21st century reduced high latitude temperature variability and increased tropical land temperature variability.

Specific comments on text

1. Lines 7-16. Abstract. This is overconfident. In particular line 12 "both robust across almost all models" actually means "Robust across 8 out of 10 models (as in Fig 3)". This would therefore be better "both shown by most models." This conclusion, as presented, is at present not really robust though the physical arguments for what is most likely to happen are reasonable.

2. Line 29: The acronym SMILEs should be written out in full here for the audience of this paper.

3. Line 20 and later quite extensively. The paper concerns itself with local to regional surface temperature variability. "Regional" is acceptable, as in the title, but it needs adding here. The nominal resolution of the temperature analysis (1x1o) is sufficiently important to understanding the text that it should be stated up front here, and not just in Methods. Its best to describe it as "nominally" 1x1o as this is not the true resolution and certainly not in HadCRUT4. So for clarity, "regional" should be added selectively in before "temperature variability" elsewhere in the paper.

4. Line 24. Although most Figures are easy to understand, Fig 1a is not. I return to this under Figures below.

5. Line 28. It would help the audience of this to explain the acronym CESM1_CAM5_LME here.

6. Line 55: This a good example of where adding "regional" after globally-averaged would make this sentence clearer.

7. Line 112. This would advantageously be stated earlier near line 28 where CESM1_CAM5_LME was first introduced.

8. Lines 127 and 128. For this audience, it would be useful to give the key references for these forcing scenarios.

9. Line 116. Fig 2a is introduced here after fig 2b (introduced in line 95).

10. Line 132. Fig S5 is In general a nice clear figure, central to the message of the paper. Fig S5 would be better as a main figure.

11. Line 137. This seems not be not the Figure really referred to.

12. Line 124. Correct about limited high latitude data, but an out of date version of HadCRUT4 is used. HadCRUT4.2 (as in Data Availability) was replaced years since by HadCRUT4.6 with slightly more Arctic data. Even better, if too late for this draft, is HadCRUT5 (Morice et al, JGR (Atmos), Dec 2020 on line). HadCRUT5 is better in the Arctic than any version of HadCRUT4 even in its "non filled" mode. HadCRUT5 also automatically incorporates HadSST4. Its not essential to change this observed temperature data in this paper but some readers might note use of old HadCRUT data. DadCRUT4.6 and HadCRUT5 are accessible from the Met Office's hadobs web site.

13. Line 151. Gives the impression that Fig 1a shows preindustrial temperature variability for these models, but pre 1850 data are not shown for individual models.

14. Line 154. This is possibly my most significant comment. There are no estimates of statistical

significance to back up the finding of robust signals of a changed pattern of temperature variability. This is the key statement in the paper. It ideally needs a statistical significance estimate for each model in Fig 3b and perhaps the 10 models collectively. All models may well not all pass such a test, but do most of them?

15. Line 154-157. This overstates the similarity of the Fig 3b results. A more nuanced statement is needed.

16. The phrase "Arctic Amplification" is best first introduced, with references, in line 181. This important idea probably deserves a second, earlier, comprehensive reference: e.g. Serreze and Francis, 2006, *Climatic Change*, 76, 241-264.

17. Line 163. The figure numbers are not in order.

18. Line 195. "Drying" vegetation cover is a bad description. Needs a more precise description.

19. Line 207. Expand on why future precipitation is modelled to increase in these African regions. Has this to do with more warming of the Northern Hemisphere tropical Atlantic and North Indian Ocean than their southern hemisphere components? Give a reference or two. If so, this general idea goes back to Folland et al 1986 and Palmer 1986, both in *Nature*.

20. Line 209. Ideally in Methods give some details of how future vegetation changes are calculated in SMILEs models. It is not enough only to give a SMILEs reference for this, as this is a crucial factor for understanding this paper.

21. Line 211-213. Explain better why variations in temperature (rather than systematic increases) should drive systematic reductions in carbon storage.

22. Lines 216 -236 are Conclusions so should have a sub-section labelled as such. This section is well written.

23. Line 242 of Methods. HadSST4 not Had4SST4

24. Line 264. I think you mean "variability is determined....."

25. Line 278. Explain what you mean by the quasi-ergodic assumption in the climate context. for this audience.

26. Line 312. This is HadISST2 sea ice. Writing HadISST2 on its own is misleading as this can refer to the SST data component. It would be good to know whether HADISS2.1 or HadISST2.2 sea ice area was used – there is a small but not very important difference in the Arctic in fairly recent years.

Specific Comments on Diagrams

1. Fig 1a: I could not understand why the year scale stopped at 1825 as model palaeodata extend to 1849 in the text but it seems to stop in 1825 in Fig 1a. Is the 20 year running averaging centred? What are the coloured vertical bars shown in the 1825-50 region? I could not see the filled dots noted in the caption. The pale yellow model information at the right is almost unreadable. In the caption, Had4SST4 should be HadSST4.

2. Fig 1b. What are the mapped dots?

3. Fig 2a-c. and caption. It would be very good to show a suitable statistic of the significance of the changes in variability distribution from Figs 2a to 2c as noted above. Again, model details shown in pale yellow are difficult to read..

4. Fig 3 is a good diagram.

5. Fig 4. Same remarks about model details shown in pale yellow. It would be clearer to expand SIA on the x axes into sea ice area. In the caption, HadISST should be described as HadISST2 sea ice area. In Fig 4a, the HadCRUT4 line is not clear – perhaps increase its thickness relative to model lines? The captions of c) and d) read oddly. Thus what is meant is the ratio of changes in

the Bowen ratio compared to its original values in c) and changes in the ratio of temperature variability. Using ratio twice in connection with Bowen ratio is correct, but can be somewhat confusing and needs careful writing.

6. Fig S1. Same remarks as above about the use of pale yellow for text. In Fig. S1e what is the grey area? The graphics at the right of Figs S1a and b need describing.

7. Fig S3. It would be useful to add the end date of each data set. On the last line of the caption, "extend" not extent..

8. Fig S5. Same remarks about use of pale yellow, but as suggested above, this could with advantage become a main diagram.

9. Fig S7. The stippling is not always clear on these small diagrams.

Prof. Chris Folland 9 Feb 2021

Author responses to reviewer comments for

Large-scale emergence of regional temperature variability by the end of the 21st century

(previously: Regional temperature variability is projected to emerge from its natural range)

D. Olonscheck, A. P. Schurer, L. Lücke, G. C. Hegerl

May 2021

Reviewer comments are shown in black.

Author responses are shown in bold red.

(line numbers reference the revised version of the document)

We thank all three reviewers for their time, the careful evaluation of our manuscript and the thoughtful and constructive comments. We are happy to hear that you all find this paper interesting and suitable for publication in Nature Communications. We have edited the manuscript to address each of the reviewer comments and feel that these changes have improved the paper.

In response to the comments, we have done the following major changes:

- 1) We changed the title to account for both when and where variability is projected to emerge.***
- 2) We used an F-test to show where projected changes in variability are significant in Fig. 4b. The largely significant changes support the robustness of our conclusions.***
- 3) We tested the dependence of the variability estimate from observations on the use of the multi-model mean to detrend observations. We find that the ratio of observed variability is robust even when we detrend the observations with single-model ensemble means, and show the results in the new Supplementary Figure S3.***
- 4) We substantially revised the previous Fig. S1 that evaluates the simulations with paleoproxies and now include it in the main text as Fig. 2. This improves the flow of the text and strengthens the paleoclimate perspective of the manuscript.***
- 5) We incorporate the previous Fig. S5 that shows the transient evolution of temperature variability change to the main Fig. 3. This complements the main message of Fig. 3.***
- 6) We now use the updated data sets HadCRUT5 and HadISST2.2 throughout the manuscript.***
- 7) We replotted most figures to account for the suggested changes.***
- 8) We added many suggested references and streamlined language throughout the manuscript.***

Reviewer #1 (Remarks to the Author):

Thank you for inviting me to review the paper: “Regional temperature variability is projected to emerge from its natural range” by Olonscheck et al.

As the authors suggest, changes in the mean state of the climate have received more attention and are better understood than changes in variability. Given that variability is an important consideration for climate extremes and adaptation, it is imperative to understand how variability might change as the planet warms. By analyzing a large set of SMILEs, observations and paleoproxies, the authors reproduce and lend additional credence to the projection that as the world warms, consistent patterns of variability change emerge across the models - variability increases over the tropical land areas and decreases over the high latitudes. The authors also briefly explore plausible mechanisms by which these changes might occur.

Recommendation: This is a well written paper. I liked reviewing it and I recommend its publication. I have a few methodological/conceptual concerns that I think would serve the authors well if addressed.

1) The title can perhaps be more specific. As it stands now, it does not seem to convey to the reader anything about when/why/where might variability emerge from its natural range. Might be good to have some of that information in the title.

Thanks for this suggestion. To account for both when and where variability is projected to emerge, we changed the title of the manuscript to “Large-scale emergence of regional temperature variability by the end of the 21st century”.

2) Introduction: There is a considerable amount of literature investigating changes in interannual variability and its emergence in a warming climate. I am listing a few at the bottom that would be useful to cite for the reader.

Thanks for these valuable references. We now included most of them in the introduction and/or results section.

3) line 23: I feel the wording is a bit awkward here. Consider rephrasing: “..which is caused by the difficulty to reliably quantify..” —> “..owing to the difficulty of reliably quantifying..”

Changed to “... due to the difficulty of reliably quantifying ...”.

4) line 24: It might be useful to briefly mention why it’s difficult to reliably estimate changes in variability using the tools mentioned

We can cleanly isolate the internal variability component because we use the large numbers of ensemble members which all contain the same forced component of variability but have different realisations of internal variability. This is not possible with single simulations or smaller

ensembles which was all that was available until very recently.

We now mention in ll. 26-29 the reason for the difficulty by changing the sentence to “The inaccuracy of disentangling the forced response and changes in internal variability with these traditional tools results in inconclusive estimates of projected change, ...”.

5) fig 1a label and elsewhere: The term “Globally averaged temperature variability” seems to imply that you are calculating variability at each grid point and then averaging across the globe. If the order of the operation is opposite, that is, you are averaging temperature across the globe first and then computing the variability, the term should be “variability of global averaged temperature”.

This is indeed an important difference. We intentionally use the term “Globally averaged temperature variability” to point to the fact that we first calculate the variability at each grid point and then average across the globe.

We now highlight this important difference in the methods section in ll. 298-301.

6) figure s1a: the computation is a bit unclear to me here. Are the 100-year periods centered around each time point? For example, to calculate the estimate of variability at year 1200, are you centering a window around that year and then taking the interannual variability in that window?

Indeed, this is how we compute the interannual variability in the 100-year periods. The years at the x-axis signify the centre of the 100-year window.

For more clarity, we substantially revised the figure and the description accordingly. We also moved this previous Fig. S1 to the main text as Fig. 2.

7) figure s1b: what do the lines to the right of the plot represent?

The bars on the right of the previous Fig. S1b represented mean and range of the standard deviation of all 100-yr periods coloured according to the respective single-forcing simulations.

We now revised the figure and use violin plots instead of lines for illustrative purposes. We further added this missing explanation to the figure caption.

8) line 61: I am a bit confused about why JJA is picked to compute variability. Since the rest of the paper deals with annual temperature, why not show the spatial distributions for annual temperature variability instead?

Indeed, we here used the variability in JJA, while we look at annual variability in the rest of the

paper. We limited the comparison of the simulated variability and N-TREND to JJA, because the tree-ring data reflect summer temperatures only. This is why we analyze summer temperatures here instead of annual means.

However, we realized that N-TREND reflects MJJA temperatures instead of just JJA temperatures. We now use MJJA for the model simulations. We explain the rationale for using summer temperatures now in ll. 63-65.

9) line 64 and figures s1c-e: I think the differences here are not negligible (e.g: the Arctic circle). I think a difference plot would show more light on the geographical patterns of differences in variability

We agree that the differences are not negligible and rephrased the sentence accordingly. We now show the difference between the models and N-TREND in the new Figure S1k-l that reveals that the models generally show larger variability than N-TREND especially over land. We further show the temperature variability for all ten PMIP models in Figure S1a-j which shows that the difference in simulated and observed variability have the same order of magnitude as inter-model differences. We added these findings to ll. 68-70.

10) figure s3 caption: ‘extent’—> ‘extend’

Changed.

11) line 158: Confidence intervals are perhaps a more refined way than range for checking whether the future temperature variability has “emerged”. Alternatively, you could consider doing an F-test at each grid point between the 2080-2099 temperatures and the sample of preindustrial temperatures.

We now show the significance of the projected emergence of temperature variability by performing an F-test at each grid point between the 2080-2099 temperatures and the preindustrial temperatures. We added stippling in Fig. 4b to show where changes in variability are significant. The regions with significance largely agree with the regions of emergence, supporting our key finding of strong increases in variability on tropical land and substantial decreases in high latitudes emerging from the natural range of interannual temperature variability.

12) line 187: wording is a bit awkward here. consider rephrasing: “..caused by Arctic amplification and hence less..” —> “..caused by Arctic amplification which leads to fewer..”

Changed as suggested.

13) line 204: “confirms” is a perhaps a strong statement here, especially since you list an exception right afterwards.

We have changed “confirms” to “suggests” to account for the mentioned exception.

14) Line 232: This is not a new result. What would be interesting here is a quantification of when exactly the emergence might happen. While questions about the emergence of the mean state have been widely discussed in the literature and researchers have come up with various ways of quantification of the mean, questions about the emergence of variability are harder to answer. A few studies that attempt to do this and are perhaps worth alluding to are: Deser et al. 2012, Yettella and England. 2018

We disagree that the finding is known that large-scale forced changes in internal interannual temperature variability will emerge from the preindustrial envelope of variability. To our knowledge, there is no literature that states that the internal temperature variability is projected to emerge by the end of the 21st century. This is because only the recently available SMILEs allow for a robust analysis of forced changes in internal variability that we present here.

However, we agree that it would be interesting to quantify when exactly the variability is projected to emerge in addition to the finding that changes emerge at the end of the century. However, the time of emergence is challenging to robustly quantify, because the time of emergence would be sensitive to a) large year-to-year fluctuations of the variability change, b) choice of how to define emergence (first year outside or at least a sequence of five years outside a past range) and c) the dependence of the exact variability change on the ensemble size. For these reasons, and because we can only consider a small set of SMILEs we decided that the quantification of when the variability emerges is presently not robustly possible.

However, we now cite both Dwyer et al., 2012 and Yettella and England, 2018 in the text. We assume that with Deser et al. 2012 the reviewer thought of Dwyer et al. 2012, because Deser et al. 2012 is about detection of changes in the mean state, while Dwyer et al. 2012 and Yettella and England, 2018 are about changes in the seasonal cycle.

15) Line 275: From a statistical standpoint, I don't think the sample standard deviation is a robust estimate of internal variability. In the limit of an infinite number of ensemble members, the sample standard deviation is indeed an unbiased estimator of internal variability (i.e., it converges to the true internal variability). In reality however, we have to settle for a small number of ensemble

members and this leads to large sampling uncertainty in the second and higher moments. This is why I think the yearly plots shown in Fig 1.a are so noisy and you had to smooth over a time window to get a more representative estimate of the true underlying internal variability. I would perhaps rephrase like the following: "...to get a more robust estimate of internal variability we smooth the sample estimate over xoxo time window...". A more detailed investigation of these nuances can be found in Yettella et al. 2018

As suggested, we added the following sentence to ll. 296-297: "We smooth the sample estimate over a 20-year time window in Figure 1a to derive more robust estimates of globally averaged temperature variability."

Generally, the robustness of the sample ensemble standard deviation to estimate internal variability indeed highly depends on the number of ensemble members available, which is shown for example in Olonscheck and Notz, 2017 and Milinski et al., 2020. We specifically mention this aspect in ll. 231-233. The multiple SMILEs with their unprecedented number of ensemble members that we use in this study are a new tool that allows for a substantially more robust estimate of forced changes in variability than ever before.

16) Line 280-285: As stated previously, confidence intervals/F-tests are a more formal way for checking whether the future temperature variability has "emerged".

See reply to comment 11)

17) Line 280-285: It took me a bit of time to figure out how exactly the stats are being computed here. One thing that could help here (and in other parts of the paper where mathematical operations are involved) is to formalize with the help of equations. That way, there will be no room for ambiguity for the reader.

As suggested, we have now formalized the statistics by adding the relevant equations on how we calculate emergence. We added the following to the method section in ll. 306-311:

"Emergence occurs if

$$\sigma_{ens}(N, 2080-2099) > \max(\sigma_{piC})$$

or

$$\sigma_{ens}(N, 2080-2099) < \min(\sigma_{piC})$$

where N is the number of ensemble simulations of each model and σ_{piC} is the standard deviation of a 100-year long segment of the preindustrial control simulation."

18) Figure 1a: Was there a reason why running means were calculated over a 20-year period?

We chose the 20-year period because we look at the 20-year period 2080-2099 in Figure 4b and for visualization purposes.

citations:

Räisänen, J., 2001: CO₂-induced climate change in CMIP2 experiments: Quantification of agreement and role of internal variability. *J. Climate*, 14, 2088–2104, [https://doi.org/10.1175/1520-0442\(2001\)014<2088:CICCIC>2.0.CO;2](https://doi.org/10.1175/1520-0442(2001)014<2088:CICCIC>2.0.CO;2).

Collins, M., and M. R. Allen, 2002: Assessing the relative roles of initial and boundary conditions in interannual to decadal climate predictability. *J. Climate*, 15, 3104–3109, [https://doi.org/10.1175/1520-0442\(2002\)015<3104:ATRROI>2.0.CO;2](https://doi.org/10.1175/1520-0442(2002)015<3104:ATRROI>2.0.CO;2).

Yettella, V., Weiss, J. B., Kay, J. E., & Pendergrass, A. G. (2018). An ensemble covariance framework for quantifying forced climate variability and its time of emergence. *Journal of Climate*, 31(10), 4117–4133. <https://doi.org/10.1175/JCLI‐D-17-0719.1>

Deser, C., Phillips, A., & Bourdette, V. (2012). Uncertainty in climate change projections: the role of internal variability. *Climate Dynamics*, 38, 527–546. <https://doi.org/10.1007/s00382-010-0977-x>

Yettella, V., and M. R. England, 2018: The role of internal variability in twenty-first-century projections of the seasonal cycle of Northern Hemisphere surface temperature. *J. Geophys. Res.*, 123, 13 149–13 167, <https://doi.org/10.1029/2018JD029066>.

Dwyer, J. G., Biasutti, M., & Sobel, A. H. (2012). Projected changes in the seasonal cycle of surface temperature. *Journal of Climate*, 25(18), 6359–6374.

Reviewer #2 (Remarks to the Author):

Review of Olonscheck et al. “Regional temperature variability is projected to emerge from its natural variability”

This study investigates future changes in surface temperature variability in various regions across the globe, using paleo-proxy records, observations and large ensemble model simulations. They find that by the end of the century, interannual temperature variability (defined as either the temporal standard deviation or the standard deviation across the large ensemble) decreases in high latitudes and is outside of the range seen in pre-industrial control runs, while in tropical land areas, interannual temperature variability increases. This is attributed to loss of sea-ice in high latitudes and changes in vegetation cover in the tropics (changing the split between sensible and latent heat).

I find this to be a very interesting and well-written study, with important implications.

1) However, I have one main concern that I think needs to be discussed. The comparison with observed internal variability is achieved by removing the forced response from the multi-model ‘SMILE’ ensemble average. The SMILE ensemble average can be assumed to be a very good estimate of the simulated forced variability, but it is unclear how alike it is to the real-world ‘true’ forced response. There may be common model biases in the forced response (that exist across all models and hence are present in the multi-model mean) which when removed from the observations either leaves part of the forced trend unremoved or introduces spurious trends in the wrong direction. If the forced trend isn’t fully removed, this could substantially bias the estimates of interannual variability.

Related to that point, the authors comment (L 78-79) that the temporal standard deviation is the more conservative estimate for interannual variability, supported by Figure S2. Figure S2 however shows the different estimates for internal variability where each model is detrended with its own forced response, therefore the separation between forced response and internal variability is ‘correct’ for that model by definition, as long as the ensemble is large enough. The same cannot be said for observations.

Unfortunately, there is no way to perfectly estimate of the real-world ‘true’ forced response, so I appreciate the difficulty of doing this and that choices have to be made, even if they aren’t perfect. However, given the importance of estimating interannual variability for this study, I think this needs to be discussed at the very least. The impacts of this de-trending choice could be tested by checking that other methods of detrending give similar results, or perhaps checking how the results would be affected by using model X’s forced response to detrend model Y (or use different models separately to detrend the observations – which will likely be worse than the MMM since the forced response of the MMM is likely to be less biased than a single model’s, but I think it would good to show some

examples of how different estimates of the forced response affect the results).

Thanks a lot for this important comment, which we fully agree with. Detrending the observations with the multimodel mean (MMM) of the SMILE means is an assumption that requires justification and proof to be the best choice. Among others, Frankcombe et al (2018) have shown that the removal of the MMM is the most defensible choice to detrend observations, as they find that the MMM outperforms single-model ensemble means where only a single realisation – such as for observations – is available.

In addition, we analysed the sensitivity to that choice by a sensitivity test detrending the observations (i.e. HadCRUT5) with each single-model forced response. We show the results in the new Fig. S3, which detrends the observations with each individual SMILE mean instead of the MMM. Results show that the ratio of observed variability shown in Fig. 1c is insensitive to the choice of the forced response that is used to detrend the observations. This result can be explained by the sufficiently large ensemble size of each SMILE to robustly represent the forced response.

Reference: Frankcombe, L., M. England, J. Kajtar, M. Mann, and B. Steinman, 2018: On the choice of ensemble mean for estimating the forced signal in the presence of internal variability. J. Climate, 31, 5681–5693, <https://doi.org/10.1175/JCLI-D-17-0662.1>.

Other than that, while some aspects of future changes in variability have been shown before (e.g. decrease in high-lat variability), the combination of paleo-proxy records, observations and large ensembles is novel and a useful contribution to our understanding of how interannual climate variability might change in the future. I would therefore support the publication of this article subject to the authors addressing my comment above. I also include a few minor comments.

Minor comments:

2) L 43: where does the value 0.47 degC come from? The authors point to Figure S1a, but that figure seems to suggest values between 0.15-0.20 degC unless I misunderstand either the text or the figure. This does need to be addressed but should be easy to fix (or clarify if I have misunderstood)

We pointed to Figure 1a, not Figure S1a. The difference is that we show globally averaged temperature variability in Figure 1a, whereas we show variability in global-mean temperature in the previous Figure S1a (now Figure 2a).

The 0.47°C come from averaging all model estimates of globally averaged temperature variability shown in Fig. 1a. We now added the average estimate of 0.44°C from observations in l. 47 for better comparison.

3) If there is space, I think the authors should consider making figure S1 part of the main manuscript as it really helps the flow of the paper and is key for understanding.

Thanks for this suggestion. For the given reasons, we decided to move the previous Figure S1 to the main part as Figure 2. Furthermore, we substantially revised the figure for more clarity and illustrative purposes.

4) L 43-44: consider mentioning the de-trending here rather than only in the caption of the Supplementary figure.

We now also mention the detrending in ll. 45-46.

5) L51-58: it isn't clear from the text whether the increased variability following volcanic eruptions in model simulations is considered realistic or not, only that PAGES2K is known to underestimate high-frequency variability, but that appears to be true throughout the period considered, not just for the volcanic part.

We revised these sentences in ll. 52-58 to more clearly mention that the variability in PAGES2K is insensitive to volcanic eruptions, suggesting a plausible magnitude of variability in the model simulations. To further support this, we added the following references:

Hartl-Meier, C. T. M. et al., 2017: Temperature Covariance in Tree Ring Reconstructions and Model Simulations Over the Past Millennium. Geophysical Research Letters, 44, 9458–9469, <https://doi.org/10.1002/2017GL073239>.

Stoffel, M. et al., 2017: Estimates of volcanic-induced cooling in the Northern Hemisphere over the past 1,500 years. Nature Geoscience, 8, 784–788 (2017). <https://doi.org/10.1038/ngeo2526>.

6) L54-58 is a bit confusing without a sentence more introducing the single forcing experiments of CESM-CAM5 LME or Figure S1. I would suggest inserting "... Driving changes in simulated globally averaged natural temperature variability..." As well as inserting "natural" in front of "greenhouse gas concentrations" L57 to emphasise that you are talking about natural variations of GHGs here (it becomes clear when reading carefully and checking the Supplementary Figure, but not all readers do that!)

We rephrased the sentence to better introduce the single forcing experiments of CESM-CAM5 LME.

7) L69 and L78-79: see major comment above regarding comparison between observed temporal variability and ensemble standard deviation in SMILES.

See response to comment 1).

8) L90: small suggestion: personally, I would say “some confidence” rather than “confidence” given the amount of available evidence, but this is up to the authors

We changed this to “some confidence” as suggested.

9) L 111-112: I’m a bit confused here as to what you mean by “long-term changes caused by natural variability”. I would not expect solar and volcanic forcing to have truly long-term effects on variability (maybe a decade or two at most), are you talking about longer time scales than that (time scales of orbital changes) here? The next sentence suggests you look at changes over 10 years, which I would not class as long term. Please clarify.

We removed the word “long-term” here because it is confusing. We now more clearly explain that we compare the historical and future changes in temperature variability to the magnitude of variability caused by natural external forcings.

10) L 173-174: “at the expense of” is a bit confusing wording, somehow suggests a causal link between the tropical changes delaying the high latitude changes

We rephrased the sentence and now say “..., in contrast to their high-latitude decreases”.

Reviewer #3 (Remarks to the Author):

General

This paper uses an extensive set of model simulations of past and future climate to investigate the likely changes in worldwide local to regional surface air temperature variability during the late 21st century compared to the last 1000 years and the recent climate. The main conclusion of the paper is that there is evidence for, and good physical reasons for, a decrease of high latitude temperature variability and an increase of tropical land temperature variability. The physical arguments are stronger than the model evidence because not all models are fully consistent in showing increased tropical temperature variability in the later 21st century. As far as they go (see below) the statistical analyses are appropriate but probably insufficient.

This topic is a suitable one for Nature Communications as it is important physically, and as mentioned in the paper, for potential impacts on tropical society. The paper needs a fair amount of detailed, largely minor, revision. There is some over confidence in the strength of the conclusions about the global pattern of future variability based on the not totally consistent model evidence. Thus there are no conclusive statistical tests on this key topic so I strongly recommend that more work be done here. Without this, the paper may still be publishable but would not be so compelling in its results. Over confidence in the conclusions is reflected in the current wording of the Abstract. At least tentatively, I recommend publication of this paper, subject to the revisions below. My recommendation would be considerably stronger if further statistical tests confirmed the statistical significance of the pattern of modelled late 21st century reduced high latitude temperature variability and increased tropical land temperature variability.

We have added more statistical significance testing to improve the robustness of results and have carefully reworded the paper where the conclusions were phrased too strongly. We hope the reviewer finds the paper now improved.

Specific comments on text

1. Lines 7-16. Abstract. This is overconfident. In particular line 12 “both robust across almost all models” actually means “Robust across 8 out of 10 models (as in Fig 3)”. This would therefore be better “both shown by most models.” This conclusion, as presented, is at present not really robust though the physical arguments for what is most likely to happen are reasonable.

We have changed the abstract to “both shown by most models” as suggested.

2. Line 29: The acronym SMILEs should be written out in full here for the audience of this paper.

The acronym SMILEs was already written out in full. However, we revised the order of words

from “single-model initial-condition large ensembles from multiple models (SMILEs)” to “single-model initial-condition large ensembles (SMILEs) from multiple models” to increase clarity.

3. Line 20 and later quite extensively. The paper concerns itself with local to regional surface temperature variability. “Regional” is acceptable, as in the title, but it needs adding here. The nominal resolution of the temperature analysis (10x10) is sufficiently important to understanding the text that it should be stated up front here, and not just in Methods. Its best to describe it as “nominally” 10x10 as this is not the true resolution and certainly not in HadCRUT4. So for clarity, “regional” should be added selectively in before “temperature variability” elsewhere in the paper.

We added the important information of the nominal spatial resolution of all data and simulations of 1°x1° to the main text in ll. 98-99. We also use the term “regional temperature variability” more frequently.

4. Line 24. Although most Figures are easy to understand, Fig 1a is not. I return to this under Figures below.

See answer below.

5. Line 28. It would help the audience of this to explain the acronym CESM1_CAM5_LME here.

We now explain the acronym CESM1-CAM5 LME in l. 33 for clarity.

6. Line 55: This a good example of where adding “regional” after globally-averaged would make this sentence clearer.

We now use the term “globally averaged regional temperature variability” here and elsewhere for more clarity.

7. Line 112. This would advantageously be stated earlier near line 28 where CESM1_CAM5_LME was first introduced.

As suggested, we moved this sentence to the introduction.

8. Lines 127 and 128. For this audience, it would be useful to give the key references for these forcing scenarios.

We have added the following key reference for the forcing scenarios here:

- van Vuuren, D. P. et al., 2011: The representative concentration pathways: an overview. Climatic Change 109, 5–31.

- Gidden, M. J., Riahi, K., Smith, S. J., Fujimori, S., Luderer, G., Kriegler, E., van Vuuren, D. P., van den Berg, M., Feng, L., Klein, D., Calvin, K., Doelman, J. C., Frank, S., Fricko, O., Harmsen, M., Hasegawa, T., Havlik, P., Hilaire, J., Hoesly, R., Horing, J., Popp, A., Stehfest, E., and Takahashi, K., 2019: Global emissions pathways under different socioeconomic scenarios for use in CMIP6: a dataset of harmonized emissions trajectories through the end of the century, Geosci. Model Dev., 12, 1443–1475, <https://doi.org/10.5194/gmd-12-1443-2019>.

9. Line 116. Fig 2a is introduced here after fig 2b (introduced in line 95).

For a more consistent order of figure panel references, we now also refer to (now) Figure 3a in l. 102.

10. Line 132. Fig S5 is In general a nice clear figure, central to the message of the paper. Fig S5 would be better as a main figure.

We agree that the previous Fig. S5 nicely illustrates the central message of the paper. We decided to incorporate the figure in the main part by adding it to Fig. 3 as panels g-i. The new panels complement the message of the previous Fig. 3 by showing the transient evolution of the temperature variability change in addition to the time slices selected for panels a-f.

11. Line 137. This seems not be not the Figure really referred to.

We apologize for this mistake. We have now corrected the figure reference to Fig. 3.

12. Line 124. Correct about limited high latitude data, but an out of date version of HadCRUT4 is used. HadCRUT4.2 (as in Data Availability) was replaced years since by HadCRUT4.6 with slightly more Arctic data. Even better, if too late for this draft, is HadCRUT5 (Morice et al, JGR (Atmos), Dec 2020 on line). HadCRUT5 is better in the Arctic than any version of HadCRUT4 even in its “non filled” mode. HadCRUT5 also automatically incorporates HadSST4. Its not essential to change this observed temperature data in this paper but some readers might note use of old HadCRUT data. HadCRUT4.6 and HadCRUT5 are accessible from the Met Office’s hadobs web site.

Thanks for this important suggestion. We decided to use the most recent data from HadCRUT5

for this paper and find very similar results with this updated data set (see Figures 1a,c, S4 and S5). Now, also HadCRUT5 defines the mask of data gaps and uncertainty.

In Fig. 1a, the use of HadCRUT5 (instead of HadCRUT4) as a mask for all observational products caused a better agreement of the observational estimates of globally averaged temperature variability. This further supports the general agreement between observed and simulated globally averaged temperature variability.

13. Line 151. Gives the impression that Fig 1a shows preindustrial temperature variability for these models, but pre 1850 data are not shown for individual models.

Figure 1a also shows preindustrial temperature variability for all models based on the variability of their preindustrial control simulations shown as filled dots with uncertainty bars. We now specifically point to the filled dots in Figure 1a for more clarity.

14. Line 154. This is possibly my most significant comment. There are no estimates of statistical significance to back up the finding of robust signals of a changed pattern of temperature variability. This is the key statement in the paper. It ideally needs a statistical significance estimate for each model in Fig 3b and perhaps the 10 models collectively. All models may well not all pass such a test, but do most of them?

We now added significance stippling based on an F-test in (now) Fig. 4b for each model and the ten models collectively. The result shows that for most regions of emergence the change in temperature variability of the period 2080-2099 compared to the preindustrial control variability is statistically significant. This is in agreement with our definition of emergence as a regional deviation of the projected temperature variability for the period 2080-2099 from the full range of past unforced temperature variability, but allows to associate a clearer statistical uncertainty to this deviation from past variability.

We further discuss the differences between the regions of emergence and regions with a statistically significant change in temperature variability from the F-test in ll. 313-317.

15. Line 154-157. This overstates the similarity of the Fig 3b results. A more nuanced statement is needed.

To account for the differences in the model results in Fig. 4b, we revised the statement and now say that “most” SMILEs agree on broad features of emergence at the end of the 21st century.

16. The phrase “Arctic Amplification” is best first introduced, with references, in line 181. This important idea probably deserves a second, earlier, comprehensive reference: e.g. Serreze and Francis, 2006, *Climatic Change*, 76, 241-264.

We added the term Arctic Amplification for more clarity and also include the suggested reference here.

17. Line 163. The figure numbers are not in order.

We rephrased the sentence to reflect the order of figure panels.

18. Line 195. “Drying” vegetation cover is a bad description. Needs a more precise description.

We agree. We have now replaced “drying vegetation cover” by the more precise description “transition to drier surface types, e.g. from tropical forests to semi-arid landscapes”.

19. Line 207. Expand on why future precipitation is modelled to increase in these African regions. Has this to do with more warming of the Northern Hemisphere tropical Atlantic and North Indian Ocean than their southern hemisphere components? Give a reference or two. If so, this general idea goes back to Folland et al 1986 and Palmer 1986, both in *Nature*.

There is no consensus in the literature about the reasons for the projected increase in precipitation in North and East Africa, with different mechanisms discussed. Therefore, we added the sentence: “There is no consensus yet on the causes for the projected increased precipitation in North and East Africa (Folland et al, 1986; Monerie et al, 2021)”, with reference to two studies that suggest different mechanisms.

20. Line 209. Ideally in Methods give some details of how how future vegetation changes are calculated in SMILEs models. It is not enough only to give a SMILEs reference for this, as this is a crucial factor for understanding this paper.

We added the subsection “Modelling of vegetation in SMILEs.” in Methods to describe how the SMILE models simulate vegetation dynamics. We also added the relevant reference there:

*Dirmeyer, P. A. et al., 2021: Projected hydroclimate changes driven by carbon dioxide trends and vegetation modeling in CMIP6. *Earth and Space Science Open Archive*, 34.
<https://doi.org/10.5291002/essoar.10506162.1>.*

21. Line 211-213. Explain better why variations in temperature (rather than systematic increases) should drive systematic reductions in carbon storage.

We decided to remove the two sentences on carbon storage, because they are not relevant for the storyline of this paper.

22. Lines 216 -236 are Conclusions so should have a sub-section labelled as such. This section is well written.

We added the sub-section label “Conclusions” here. Thanks for the nice words!

23. Line 242 of Methods. HadSST4 not Had4SST4

Thanks for this note. We reconsidered how to best name the data set by Cowtan & Way (2014). We decided to use the name “HadCRUT4-CW” instead of Had4SST4 and mention in the methods that this refers to the krigged version of HadCRUT4 from Cowtan & Way (2014). We also added the more recent reference Cowtan et al. (2015). We changed this here and everywhere else in the manuscript.

Reference: Cowtan, K. et al., 2015: Robust comparison of climate models with observations using blended land air and ocean sea surface temperatures. Geophysical Research Letters, 6526–6534. <https://doi.org/10.1002/2015GL064888>.

24. Line 264. I think you mean “variability is determined.....”

Thanks a lot. We changed “variance” to “variability”.

25. Line 278. Explain what you mean by the quasi-ergodic assumption in the climate context. for this audience.

We describe the quasi-ergodic assumption by adding the phrase “which states that the variance of one sequence of events over time equals the ensemble variance at a given time” for more clarity.

26. Line 312. This is HadISST2 sea ice. Writing HadISST2 on its own is misleading as this can refer to the SST data component. It would be good to know whether HADISS2.1 or HadISST2.2 sea ice area was used – there is a small but not very important difference in the Arctic in fairly recent years.

For sea ice area, we use version HadISST2.2.0.0, which we now specify in Methods.

Specific Comments on Diagrams

1. Fig 1a: I could not understand why the year scale stopped at 1825 as model palaeodata extend to 1849 in the text but it seems to stop in 1825 in Fig 1a. Is the 20 year running averaging centred? What are the coloured vertical bars shown in the 1825-50 region? I could not see the filled dots noted in the caption. The pale yellow model information at the right is almost unreadable. In the caption, Had4SST4 should be HadSST4.

The time scale in Fig. 1a stops at 1850, not 1825. The time labelling is in steps of 50 years. We now mention in the figure caption that the 20-year running average is centred. The coloured filled dots and vertical bars show the standard deviation across the preindustrial control simulations, and the range of preindustrial temperature variability derived from all standard deviations across consecutive 100-year running means from each preindustrial control simulation, as mentioned in the caption. We have rephrased the figure caption for more clarity. We corrected to HadCRUT4-CW, and pale yellow to “gold”.

2. Fig 1b. What are the mapped dots?

We now explain the stippling in the caption of Fig. 1: “Stippling marks significant changes in temperature variability at a 5% level based on an F-test.”

3. Fig 2a-c. and caption. It would be very good to show a suitable statistic of the significance of the changes in variability distribution from Figs 2a to 2c as noted above. Again, model details shown in pale yellow are difficult to read.

We agree that a suitable statistic would be beneficial here and tried to provide one already in the initial manuscript. However, this is not possible with the available data because CESM1-CAM5 LME used for (now) Fig. 3a is a different model than all of the ten SMILE models used for (now) Fig. 3b, c.

Therefore, we prefer to just show the visual comparison between Fig. 3a and Fig. 3b, c that highlights the fact that the historical and future changes in the distribution of temperature variability are unlike the range of variability in the last millennium as simulated by CESM1-CAM5 LME, the only last millennium ensemble that we currently have.

The proper statistics for changes in regional temperature variability is provided in Fig. 4b.

4. Fig 3 is a good diagram.

Thanks for these encouraging words.

5. Fig 4. Same remarks about model details shown in pale yellow. It would be clearer to expand SIA on the x axes into sea ice area. In the caption, HadISST should be described as HadISST2 sea ice area. In Fig 4a, the HadCRUT4 line is not clear – perhaps increase its thickness relative to model lines? The captions of c) and d) read oddly. Thus what is meant is the ratio of changes in the Bowen ratio compared to its original values in c) and changes in the ratio of temperature variability. Using ratio twice in connection with Bowen ratio is correct, but can be somewhat confusing and needs careful writing.

Thanks for this comment. We have rephrased the figure caption accordingly and revised the figure. Please note that we erroneously used HadISST1 sea ice area instead for HadISST2 sea ice area in Fig. 5a. We have now updated to HadISST2.2, which has a generally larger sea ice area than HadISST1. However, this does not impact the conclusions.

We also changed “pale yellow” to “gold”.

Please further note that in Fig. 5a, we had not used HadCRUT4 but NOAA-20C for the observational estimate of Arctic mean temperatures because of its complete data coverage. We tried to use the infilled HadCRUT5 instead, but they are only anomalies, not absolute values.

We changed HadCRUT4 to NOAA-20C in the caption of Fig. 5.

6. Fig S1. Same remarks as above about the use of pale yellow for text. In Fig. S1e what is the grey area? The graphics at the right of Figs S1a and b need describing.

We added here that the grey areas in e indicate land with no data. We substantially revised the figure and now also better describe the right-hand graphics. See reply to reviewer #1 comment 7).

Please note that the previous Figure S1 is now Figure 2 in the main text.

7. Fig S3. It would be useful to add the end date of each data set. On the last line of the caption, “extend” not extent..

We added the end date of each data set and rephrased the caption accordingly.

8. Fig S5. Same remarks about use of pale yellow, but as suggested above, this could with advantage

become a main diagram.

See reply to reviewer #3 comment 10). We have changed the color “pale yellow” to “gold” throughout the manuscript.

9. Fig S7. The stippling is not always clear on these small diagrams.

Sorry for the small diagrams, but a better resolution of the stippling would require a much larger figure size.

Prof. Chris Folland 9 Feb 2021

REVIEWER COMMENTS

Reviewer #1 (Remarks to the Author):

I thank the authors for comprehensively addressing the questions posed by the reviewers. I recommend publication.

Reviewer #2 (Remarks to the Author):

I have read the revised version of Olonscheck et al. and am satisfied overall with their response to my comments. I think this is a valuable contribution and an interesting paper, but the paper would benefit from some further clarifications (mainly just re-wording some sentences).

One issue I had not previously picked up on is the choice to spatially average the local standard deviations, instead of calculating the standard deviation of the global mean. On the wording: "the globally averaged regional estimates" is somewhat confusing. Why not "the global average of local standard deviations" or something similar? That would be much clearer. However, I am also not quite sure what the physical interpretation of globally averaging regional standard deviations is – the variability of the global mean surface temperature is much more straightforward to interpret. Perhaps the authors could comment on this choice? I also wonder if it would be more accurate to average the variances, then take the square root of the averaged variance? I am not sure if this is necessary, so just bringing it up for the authors to consider.

I would recommend that the manuscript be accepted for publication following one more round of revisions to clarify some of the language and the above point. It is at times hard to follow the description of what has been done (methods) so I have tried to point out the areas that could do with some clarification. Overall, I would suggest the authors carefully check if sentences can be simplified, split up or shortened. However, these revisions are fairly easy to do and so I think the paper should be published after minor revisions.

Minor comments:

Title: I would suggest adding the work the title to "Large-scale emergence of regional changes in interannual (?) temperature variability by the end of the 21st century" as it is more informative (i.e. adding the words "changes" and "interannual" (if that is indeed correct)

L 20: perhaps relatively well understood? There are still many open questions too, but I get your meaning!

L26: are they all inaccurate? Perhaps challenges would be a better word here?

L61: would suggest adding "natural variations in" greenhouse gas concentrations to avoid confusion

L92ff: the wording of the sentence is confusing. You write: "possible increases in observed variability [...] would be supported by temporarily increased variability [...]. I think you mean "could be explained by"?"

L94: "... ,which is supported by": consider starting a new sentence here.

L120 would suggest inserting regional in the sentence: we investigate the patterns of regional variability change.

L 125-126: "naturally forced 10-year averaged change in internal temperature variability": what

exactly are you doing here? Are you talking about the temporal standard deviation of 10-year means, or the average standard deviation of multiple standard deviations of 10-year chunks? Why do you use 10 years here and 50 years earlier?

Related caption Figure 3: "Range of naturally forced 10-year averaged globally averaged": Again it isn't clear what is averaged and in what order, what does the 10 years refer to. Running means? 10 year chunks and standard deviation calculated on yearly values?

L129-130: be careful with language here: "uniform pattern of internal temperature variability change caused by natural external forcings": do you really mean that the intrinsic characteristics of internal variability changes due to natural external forcings? (also earlier, lines 125-126). Or do you simply mean the range of temperature variability due to external forcings?

L 136: again you are implying that natural external forcings have changed in response to external forcings, is this really what you mean?

L 153-156: "This is not yet evident in observations, in accordance with the mechanisms of temperature variability change of first increasing variability with a more seasonal ice cover in higher latitudes, accompanied by decreasing variability towards the equator that encompasses all the polar regions when the sea ice is gone (see below)". This sentence is confusing, please consider re-writing. You are first talking about high latitude changes, where the variability first increases. The part about "decreasing variability towards the equator that encompasses all the polar regions when the sea ice is gone" is unclear, why are you talking about changes towards the equator here? Also not sure what "see below" refers to. Perhaps also add which figure supports this discussion?

L 190-191: please specify latitudinally contrasting pattern, which I think is what you mean? The way the sentence is written implies the contrast is between the present-day and the future

L 200: please spell out which mechanism is the former: do you mean the larger ocean heat capacity? Or the decreased meridional temperature gradient?

Methods, L 270-271: what kind of re-gridding?

Methods, L 305: overlapping or non-overlapping?

Methods, L 307-309: the equation suggest that you calculate 20-year means. Is the piControl similarly smoothed? General comment for the entire paper: please be really clear when/where you perform any temporal smoothing/ averaging

Methods, L 315-317: sentence structure, move "for models with long preindustrial control simulations, i.e. MPI-ESM-LR," to the beginning or end of sentence to improve readability

Reviewer #3 (Remarks to the Author):

This paper has had considerable attention since its first version and takes account of the many reviewers' comments. It is well written with good diagrams and now acceptable for publication, subject to the minor comments below. It is definitely suitable for Nature Communications and has about the appropriate balance of text, diagrams and Supporting Information, subject to the minor comments below that will only have a small effect on the final manuscript.

My main comment is that earlier work, partly by this group and published in Holmes et al (2016) (reference 8), claimed to see an increase in future temperature variability north of 80N in DJF, but saw the reverse in JJA. An overall increase in variability in this region is faintly visible in the all seasons results of Fig 4. Moreover, Holmes et al claimed their results were "Robust" in their paper

title, though their paper uses much less data than here and has a somewhat different approach. Accordingly, the authors should comment on the above finding in the light of their new results. Do the authors make an insufficient seasonal breakdown of their high latitude results? In Holmes et al separate winter and summer results are prominent.

Minor comments

Line 43:

Although in the rest of the paper the distinction between global average temperature variability and the global average of grid point temperature variability is clear, it needs to be made more explicit at this early point. So "local" or "grid point" should be added in front of "near-surface"

Line 58 or thereabouts:

A recent paper by Chylek et al (2020) in GRL shows that CMIP5 models overestimate volcanic cooling and this is likely to be true here. So the contribution of volcanoes in Figs 2a, 2b to modelled temperature variability is likely to be too large, much as suggested by the paleoclimate proxies, though these might have an opposite fault.

Line 88, Fig 1c and the section starting at Line 100 on Inconsistent Global Mean Change:

A problem not explicitly mentioned that likely affects Fig 1c and the above section, though not crucial for the key conclusions of the paper, is worth commenting on. Natural multidecadal changes in ENSO variability from the 1870s were highlighted by Kestin et al, 1998, *J. Climate*. This may explain the lack of tropical ocean temperature variability consistency found between Fig 1b, 1c and, at least partly, in future global mean grid point temperature variability. It is quite likely that tropical Pacific grid point temperature variability in the period about 1876-1919 exceeded that in 1920-1969, as suggested by the ENSO variability in Kestin et al. The earlier period may in fact be quite like 1970-2019. This likelihood can be approximately checked using eigenvector reconstructed HadISST1 SSTs (tropical Pacific pre 1876 in that data set are not reliable). Thus are different future model tropical Pacific temperature variability behaviours over periods of 50 or more years ahead an important cause of the disagreements seen in Fig 3g?

Fig 1 caption.

Clarity would be improved if much of a sentence in the caption of Fig S2 was added: "The change in variability is determined as the ratio between the periods 1970-2019 and 1920-1969".

References

Chylek, P., C. Folland, D J. Klett, and M. K. Dubey, 2020: CMIP5 climate models overestimate cooling due to volcanic aerosols. *Geophysical Research Letters*, 47, e2020GL087047 doi.10.1029/2020GL087047

Kestin, T.S., D.J. Karoly, J-I. Yano and N.A. Rayner, 1998: Time-Frequency variability of ENSO and stochastic simulations. *J. Climate*, 11, 2258-2272.

Chris Folland 9 June 2021

Author responses to the second round of reviewer comments for

Large-scale emergence of regional changes in year-to-year temperature variability by the end of the 21st century

(previously: Large-scale emergence of regional temperature variability by the end of the 21st century)

D. Olonscheck, A. P. Schurer, L. Lücke, G. C. Hegerl

July 2021

Reviewer comments are shown in black.

Author responses are shown in bold red.

(line numbers reference the newly revised version of the document)

We thank all three reviewers for their time to continue reviewing this paper and for constructive and supportive comments. We are happy to hear that you are largely satisfied with the substantial revisions of our manuscript and that you consider the manuscript ready for publication after minor revisions. We have edited the manuscript to address each of the reviewer comments and feel that these changes further increase the clarity of the paper.

Reviewer #1 (Remarks to the Author):

We thank reviewer #1 for her/his support to improve our manuscript and for this positive evaluation.

I thank the authors for comprehensively addressing the questions posed by the reviewers. I recommend publication.

Reviewer #2 (Remarks to the Author):

We thank reviewer #2 for the constructive and supportive comments that helped to further improve the manuscript. Thanks for your valuable time and ideas. We carefully considered all the comments and took most of the improvements as suggested.

I have read the revised version of Olonscheck et al. and am satisfied overall with their response to my comments. I think this is a valuable contribution and an interesting paper, but the paper would benefit from some further clarifications (mainly just re-wording some sentences).

1) One issue I had not previously picked up on is the choice to spatially average the local standard deviations, instead of calculating the standard deviation of the global mean. On the wording: “the globally averaged regional estimates” is somewhat confusing. Why not “the global average of local standard deviations” or something similar? That would be much clearer.

We changed “globally averaged regional estimates” to “global average of local standard deviations” as suggested.

However, I am also not quite sure what the physical interpretation of globally averaging regional standard deviations is – the variability of the global mean surface temperature is much more straightforward to interpret. Perhaps the authors could comment on this choice?

We agree that the community is much more used to the variability in global mean surface temperature than in globally averaged regional standard deviations. However, people cannot experience variability in global mean surface temperature, but local temperature variability. For

this reason, we here show global patterns of local temperature variability and their change as much as possible but start with globally averaged regional standard deviations to highlight global mean differences between SMILEs. These globally averaged regional standard deviations result in much larger variability estimates than the variability in global mean surface temperature, because first globally averaging suppresses the local differences which is not what we are actually interested in.

I also wonder if it would be more accurate to average the variances, then take the square root of the averaged variance? I am not sure if this is necessary, so just bringing it up for the authors to consider.

Thanks for this note. Indeed, there are different options how to calculate globally averaged regional temperature variability. We decided to use the global average of ensemble standard deviations for easier comparison to the ensemble standard deviations used to show the patterns of temperature variability change. Because we mainly show changes in variability in this paper, the choice of using ensemble standard deviation or ensemble variance for estimating variability is not of key relevance.

I would recommend that the manuscript be accepted for publication following one more round of revisions to clarify some of the language and the above point. It is at times hard to follow the description of what has been done (methods) so I have tried to point out the areas that could do with some clarification. Overall, I would suggest the authors carefully check if sentences can be simplified, split up or shortened. However, these revisions are fairly easy to do and so I think the paper should be published after minor revisions.

We rephrased and added some description in the method section for more clarity. We also simplified some sentences where possible.

Minor comments:

2) Title: I would suggest adding the work the title to “Large-scale emergence of regional changes in interannual (?) temperature variability by the end of the 21st century” as it is more informative (i.e. adding the words "changes" and "interannual" (if that is indeed correct)

As suggested, we changed the title for more clarity to “Large-scale emergence of regional changes in year-to-year temperature variability by the end of the 21st century”. We decided to use the term “year-to-year” instead of “interannual” to account for the broad readership of Nature Communications.

3) L 20: perhaps relatively well understood? There are still many open questions too, but I get your meaning!

Changed as suggested.

4) L26: are they all inaccurate? Perhaps challenges would be a better word here?

Changed to “challenge” as suggested.

5) L61: would suggest adding “natural variations in” greenhouse gas concentrations to avoid confusion

Changed as suggested.

6) L92ff: the wording of the sentence is confusing. You write: “possible increases in observed variability [...] would be supported by temporarily increased variability [...]”. I think you mean “could be explained by”?

Changed as suggested.

7) L94: "... ,which is supported by": consider starting a new sentence here.

We here started a new sentence.

8) L120 would suggest inserting regional in the sentence: we investigate the patterns of regional variability change.

We here added the word “regional” as suggested. We added the word “regional” in many other places as well to be consistent and to avoid confusion.

9) L 125-126: “naturally forced 10-year averaged change in internal temperature variability”: what exactly are you doing here? Are you talking about the temporal standard deviation of 10-year means, or the average standard deviation of multiple standard deviations of 10-year chunks? Why do you use 10 years here and 50 years earlier?

Sorry for the confusion! We mean the average standard deviation of multiple standard deviations of 10-year chunks. Specifically, we estimate the internal variability by calculating the sample ensemble standard deviation for each year across each model’s ensemble simulations. We then average all consecutive 10-year chunks of ensemble standard deviations to smooth the variability estimate. We do this 10-year smoothing of standard deviations for CESM1-CAM5-LME because we do exactly the same for the 10-year periods 2010-2019 and 2090-2099 that we compare with the results from CESM1-CAM5-LME. We do this 10-year averaging of ensemble standard deviations for the periods 2010-2019 and 2090-2099 for a more robust and meaningful comparison.

We use the 50 years only to estimate the observed variability, where we are limited to a single record. This requires to calculate the standard deviation over time. For the most reliable result, we use the full 100 years from 1920-2019 and bisect them into two 50 years to estimate the observed change in temperature variability as the ratio between the two periods.

We explicitly mention the two different approaches in the method section in ll. 284-304 and revised the description of the temporal averaging of ensemble standard deviations in ll. 304-308 accordingly.

For more clarity, we now reordered the above mentioned term to “naturally forced change in 10-year averaged internal temperature variability” here and elsewhere in the manuscript. We now also refer to the method section in the main text in l. 132.

10) Related caption Figure 3: “Range of naturally forced 10-year averaged globally averaged”: Again it isn’t clear what is averaged and in what order, what does the 10 years refer to. Running means? 10 year chunks and standard deviation calculated on yearly values?

See reply to comment 9).

11) L129-130: be careful with language here: “uniform pattern of internal temperature variability change caused by natural external forcings”: do you really mean that the intrinsic characteristics of internal variability changes due to natural external forcings? (also earlier, lines 125-126). Or do you simply mean the range of temperature variability due to external forcings?

Yes, we think that natural external forcings, such as volcanic eruptions, do change the intrinsic characteristics of internal variability, causing the range of internal temperature variability change shown in Figure 3a. However, this impact is relatively small compared to changes in internal variability caused by anthropogenic forcing, as shown in Figure 3.

12) L 136: again you are implying that natural external forcings have changed in response to external forcings, is this really what you mean?

Sorry for the confusion. No, we don't mean that external forcings change natural external forcings. The meaning of the sentence in ll. 134-138 is that the impact of anthropogenic forcing on internal temperature variability is larger than the impact of natural external forcings on internal temperature variability. We considered re-writing this sentence, but decided to keep it as is.

13) L 153-156: "This is not yet evident in observations, in accordance with the mechanisms of temperature variability change of first increasing variability with a more seasonal ice cover in higher latitudes, accompanied by decreasing variability towards the equator that encompasses all the polar regions when the sea ice is gone (see below)". This sentence is confusing, please consider re-writing. You are first talking about high latitude changes, where the variability first increases. The part about "decreasing variability towards the equator that encompasses all the polar regions when the sea ice is gone" is unclear, why are you talking about changes towards the equator here? Also not sure what "see below" refers to. Perhaps also add which figure supports this discussion?

We agree. We re-wrote this sentence for more clarity. It now reads: "This is not yet evident in observations (Figure 1c), in accordance with the mechanisms of temperature variability change of first increasing variability with a more seasonal ice cover in higher latitudes, accompanied by decreasing variability in regions with open ocean all year long that encompasses all the polar regions when the sea ice is gone (Holmes et al. 2016) (compare Figure 4a and Section "Mechanisms of variability change")."

We replaced the term "see below" by "compare Section "Mechanisms of variability change" ". We furthermore refer to the paper by Holmes et al. 2016 and the Figures 1c and 4a that support this discussion.

14) L 190-191: please specify latitudinally contrasting pattern, which I think is what you mean? The way the sentence is written implies the contrast is between the present-day and the future

We agree. We added the word "latitudinally" to avoid the wrong meaning.

15) L 200: please spell out which mechanism is the former: do you mean the larger ocean heat capacity? Or the decreased meridional temperature gradient?

We specified "the former" by naming the decreased meridional temperature gradient again.

16) Methods, L 270-271: what kind of re-gridding?

We specified the re-gridding method by adding the words "by bilinear interpolation".

17) Methods, L 305: overlapping or non-overlapping?

We specified here and elsewhere that the consecutive 100-year periods are overlapping.

18) Methods, L 307-309: the equation suggest that you calculate 20-year means. Is the piControl similarly smoothed? General comment for the entire paper: please be really clear when/where you perform any temporal smoothing/ averaging

We agree that we need to be more explicit here. For the period 2080-2099, we calculate the ensemble standard deviation for each individual year across each SMILE, and then average the 20 ensemble standard deviations. This is not a temporal smoothing of the temperature timeseries, but an averaging of 20 consecutive estimates of temperature variability to derive more robust results. We have to treat the piC simulation differently, because this is just one simulation. This is why we use all consecutive overlapping 100-year periods to calculate the temporal standard deviation from each of the 900 estimates in case of a 1000 year long piC simulation. We do so to determine the maximum range of variability estimates from each piC simulation. See also our reply to comment 9).

We in part rephrased the description in the methods section in ll. 284-308 for more clarity on what exactly we compare. We furthermore added the following sentence to ll. 321-323: "The ensemble standard deviations derived for each year are averaged for the 20-year time window 2080-2099 for robust results of the projected end-of-century temperature variability."

19) Methods, L 315-317: sentence structure, move "for models with long preindustrial control simulations, i.e. MPI-ESM-LR," to the beginning or end of sentence to improve readability

Changed as suggested.

Reviewer #3 (Remarks to the Author):

We thank reviewer #3 for the encouraging evaluation of our manuscript and the constructive and supportive comments that helped to further improve the manuscript. Thanks for your valuable time and ideas. We carefully considered all the comments and took most of the improvements as suggested.

This paper has had considerable attention since its first version and takes account of the many reviewers' comments. It is well written with good diagrams and now acceptable for publication, subject to the minor comments below. It is definitely suitable for Nature Communications and has about the appropriate balance of text, diagrams and Supporting Information, subject to the minor comments below that will only have a small effect on the final manuscript.

20) My main comment is that earlier work, partly by this group and published in Holmes et al (2016) (reference 8), claimed to see an increase in future temperature variability north of 80N in DJF, but saw the reverse in JJA. An overall increase in variability in this region is faintly visible in the all seasons results of Fig 4. Moreover, Holmes et al claimed their results were "Robust" in their paper title, though their paper uses much less data than here and has a somewhat different approach. Accordingly, the authors should comment on the above finding in the light of their new results. Do the authors make an insufficient seasonal breakdown of their high latitude results? In Holmes et al separate winter and summer results are prominent.

Our results agree with the seasonal breakdown in Holmes et al. 2016 which is explained by the evolution of sea ice loss in high latitudes. Under global warming, there is first an increase in temperature variability in DJF because the winter sea ice area becomes much more variable with more and more regions that have sea ice in one year but not another, causing the interannual temperature variability to increase. Then, as soon as the sea ice area disappears in all years (as is first the case in JJA), the temperature variability decreases substantially because of the higher heat capacity of open water and the reduced meridional temperature gradient due to Arctic Amplification. This seasonal breakdown is analogous to the yearly-averaged evolution over time that we show in Figure 4a with the Hovmoeller plots. This is seen in all models and the timing depends on when the models start to lose summer and winter sea ice. We describe this in ll. 94-100. Therefore, we decided not to investigate seasonal differences in more detail, as this is already done by Holmes et al. 2016.

In response to your comment and the comment 13) by reviewer #2, we re-wrote the sentences in ll. 157-161 to account for the analogy between our results and the seasonal breakdown by Holmes et al. 2016, which we reference there.

Minor comments

21) Line 43:

Although in the rest of the paper the distinction between global average temperature variability and the global average of grid point temperature variability is clear, it needs to be made more explicit at this early point. So "local" or "grid point" should be added in front of "near-surface"

We agree. We added "local" here. We furthermore changed "globally averaged temperature variability" to "globally averaged regional temperature variability" throughout the manuscript to avoid confusion.

22) Line 58 or thereabouts:

A recent paper by Chylek et al (2020) in GRL shows that CMIP5 models overestimate volcanic cooling and this is likely to be true here. So the contribution of volcanoes in Figs 2a, 2b to modelled temperature variability is likely to be too large, much as suggested by the paleoclimate proxies, though these might have an opposite fault.

Thanks for pointing us to the recent paper by Chylek et al (2020). However, we decided to not mention its key result that CMIP5 models overestimate volcanic cooling in our manuscript, because to us it remains unclear how much the overestimated cooling impacts temperature

variability.

23) Line 88, Fig 1c and the section starting at Line 100 on Inconsistent Global Mean Change: A problem not explicitly mentioned that likely affects Fig 1c and the above section, though not crucial for the key conclusions of the paper, is worth commenting on. Natural multidecadal changes in ENSO variability from the 1870s were highlighted by Kestin et al, 1998, J. Climate. This may explain the lack of tropical ocean temperature variability consistency found between Fig 1b, 1c and, at least partly, in future global mean grid point temperature variability. It is quite likely that tropical Pacific grid point temperature variability in the period about 1876-1919 exceeded that in 1920-1969, as suggested by the ENSO variability in Kestin et al. The earlier period may in fact be quite like 1970-2019. This likelihood can be approximately checked using eigenvector reconstructed HadISST1 SSTs (tropical Pacific pre 1876 in that data set are not reliable). Thus are different future model tropical Pacific temperature variability behaviours over periods of 50 or more years ahead an important cause of the disagreements seen in Fig 3g?

Thanks for this valuable suggestion. To account for the multi-decadal changes in ENSO variability, we added the following sentence to ll. 92-94:

“The observed increase in tropical Pacific temperature variability in the period 1970-2019 compared to 1920-1969 might be caused by natural multi-decadal changes in ENSO variability (Kestin et al. 1998).”

We furthermore added this potential cause for the disagreements in Figure 3g to ll. 165-168:

“The different model responses in the tropical Pacific may be caused by strong multi-decadal changes in ENSO variability (Kestin et al. 1998) and support ...”

24) Fig 1 caption.

Clarity would be improved if much of a sentence in the caption of Fig S2 was added: “The change in variability is determined as the ratio between the periods 1970-2019 and 1920-1969”.

Thanks for this note! We changed the caption of Figure 1 as suggested.

References

Chylek, P., C.. Folland, D J. Klett, and M. K. Dubey, 2020: CMIP5 climate models overestimate cooling due to volcanic aerosols. Geophysical Research Letters,47, e2020GL087047
doi.10.1029/2020GL087047

Kestin, T.S., D.J. Karoly, J-I. Yano and N.A. Rayner, 1998: Time–Frequency variability of ENSO and stochastic simulations. J. Climate, 11, 2258-2272.

Chris Folland 9 June 2021

REVIEWER COMMENTS

Reviewer #2 (Remarks to the Author):

Thanks to the authors for addressing the comments raised in my previous review. The paper is now much clearer.

I still think the choice of globally averaging local standard deviations is not the best one, for the reasons I will briefly outline below. However, it is a very interesting paper with new conclusions that don't strongly depend on this choice and I think it should be published. I would not want this difference of opinion to stand in the way of publication, but nevertheless want to point out why I think this choice isn't great. My suggestion is that the authors be given the chance to consider the arguments outlined below, and if they then still prefer to use globally averaged local standard deviations, then the paper can be published as-is.

The authors write:

"We agree that the community is much more used to the variability in global mean surface temperature than in globally averaged regional standard deviations. However, people cannot experience variability in global mean surface temperature, but local temperature variability. For this reason, we here show global patterns of local temperature variability and their change as much as possible but start with globally averaged regional standard deviations to highlight global mean differences between SMILEs. These globally averaged regional standard deviations result in much larger variability estimates than the variability in global mean surface temperature, because first globally averaging suppresses the local differences which is not what we are actually interested in."

The statement "people cannot experience variability in global mean surface temperature" is certainly true, but equally applies, if not more so, to globally averaged standard deviations. They note that globally averaging temperature first suppresses local variability and that the global average of local standard deviations is larger. I would argue that their metric isn't a better indicator of local variability, instead, it heavily favors regions with large variability, e.g. the extra-tropics, where variability is much larger than in the tropics, for instance. This is likely why these estimates of variability are much larger. The standard deviation of the global mean on the other hand weighs all regions equally, but it is true that it suppresses local variability (though that can be said of any type of global average, by definition). For these reasons, in my opinion, the global average of local standard deviations is not a good representation of variability across the globe and not very well suited to compare observations and different SMILEs, since it will weigh certain regions of the globe much more than others. This could potentially be even more of an issue in observations if there are areas with missing data.

Small clarification re previous round of reviews: I was not suggesting the authors use the ensemble variance, but was saying that statistically, averaging the variances and then taking the square root is a more accurate estimate of the average standard deviation than averaging the standard deviations, though I doubt it would make a big difference in practice, so am not overly concerned about this point.

Additional comment: I thank the authors about the clarification regarding temporal averaging and order of operations in the Methods section "Emergence" (L323-325 in tracked changes). The text is now much clearer and I agree that what they have done makes physical sense. I think, though, that writing the equation as $\sigma_{\text{ens}}(N, \text{avg}2080-2099)$, i.e. with the average 2080-2099 inside the brackets suggests that the 20-year average is performed before taking the standard deviation, i.e. that they take the standard deviation of 20-year means. I don't think this is what they have done, so I would suggest carefully checking the notation of the equation. I think it should be something like $\text{avg}(\sigma_{\text{ens}}, 2080..2099)(N, t)$

Reviewer #3 (Remarks to the Author):

General

The authors have made a very good responses to the reviews so I am happy, subject to a small editorial comment below, to see this paper published as it is. The point not accepted for action about the over-sensitivity of the models to volcanic eruptions in the past would I think make modelled local temperature variability somewhat too great qualitatively. But I agree this cannot be quantified from existing data and this problem does not affect the key results of the paper.

Small Editorial point

In Fig. 4b, right of top diagram, what does the C mean in πC ? I expected π .

Chris Folland Aug 4 2021

Author responses to the third round of reviewer comments for

Large-scale emergence of regional changes in year-to-year temperature variability by the end of the 21st century

D. Olonscheck, A. P. Schurer, L. Lücke, G. C. Hegerl

October 2021

Reviewer comments are shown in black.

Author responses are shown in bold red.

(line numbers reference the newly revised version of the document)

Reviewer #2 (Remarks to the Author):

We thank reviewer #2 for the continuous supportive comments that helped to further improve the manuscript. We carefully considered the remaining comments and revised the manuscript accordingly.

Thanks to the authors for addressing the comments raised in my previous review. The paper is now much clearer.

I still think the choice of globally averaging local standard deviations is not the best one, for the reasons I will briefly outline below. However, it is a very interesting paper with new conclusions that don't strongly depend on this choice and I think it should be published. I would not want this difference of opinion to stand in the way of publication, but nevertheless want to point out why I think this choice isn't great. My suggestion is that the authors be given the chance to consider the arguments outlined below, and if they then still prefer to use globally averaged local standard deviations, then the paper can be published as-is.

The authors write:

"We agree that the community is much more used to the variability in global mean surface temperature than in globally averaged regional standard deviations. However, people cannot experience variability in global mean surface temperature, but local temperature variability. For this reason, we here show global patterns of local temperature variability and their change as much as possible but start with globally averaged regional standard deviations to highlight global mean differences between SMILEs. These globally averaged regional standard deviations result in much larger variability estimates than the variability in global mean surface temperature, because first globally averaging suppresses the local differences which is not what we are actually interested in."

The statement "people cannot experience variability in global mean surface temperature" is certainly true, but equally applies, if not more so, to globally averaged standard deviations. They note that globally averaging temperature first suppresses local variability and that the global average of local standard deviations is larger. I would argue that their metric isn't a better indicator of local variability, instead, it heavily favors regions with large variability, e.g. the extra-tropics, where

variability is much larger than in the tropics, for instance. This is likely why these estimates of variability are much larger. The standard deviation of the global mean on the other hand weighs all regions equally, but it is true that it suppresses local variability (though that can be said of any type of global average, by definition). For these reasons, in my opinion, the global average of local standard deviations is not a good representation of variability across the globe and not very well suited to compare observations and different SMILEs, since it will weigh certain regions of the globe much more than others. This could potentially be even more of an issue in observations if there are areas with missing data.

Thanks a lot for further clarifying your concern with our choice of globally averaging local standard deviations. As suggested, we have now redone Figure 1a with the standard deviation of global mean temperature. We added this alternative approach as Figure S1. We agree that the alternative approach of using the standard deviation of global mean temperature provides additional insights to better understand the evolution of global mean temperature.

In line with your arguments, more models show no change or increases in variability of global mean temperature rather than decreases than we find for the globally averaged regional standard deviations. However, our key conclusions / motivation that 1) the magnitude of variability is consistent across the last millennium, piC, historical simulations and observations, and 2) the direction of change is inconsistent across the SMILEs, remain.

We now added Figure S1 to the manuscript and refer to it in l. 50, l. 111 and the caption of Figure 1. Specifically, we now discuss the two approaches in the Methods section in ll. 308-317. We decided to keep the globally averaged regional standard deviations to be consistent with the regional changes in variability which are the focus of the paper and link to the underlying mechanisms. Also, several fundamental publications emphasize the need to globally aggregate to detect climate change early among large regional variability (e.g. Hasselmann, 1979) while recent work on emergence of climate change emphasizes that people experience climate change in regional, not global variability (see IPCC AR6, Chapter 1; note that emergence is expected later than detection of change in global mean temperature fundamentally). Hence our rationale is well supported by the scientific literature.

Reference: Hasselmann, K., 1979: On the signal-to-noise problem in atmospheric response studies. Meteorology over the Tropical Oceans, D. B. Shaw, Ed., Roy. Meteor. Soc., 201–259.

Small clarification re previous round of reviews: I was not suggesting the authors use the ensemble variance, but was saying that statistically, averaging the variances and then taking the square root is a more accurate estimate of the average standard deviation than averaging the standard deviations, though I doubt it would make a big difference in practice, so am not overly concerned about this point.

Additional comment: I thank the authors about the clarification regarding temporal averaging and order of operations in the Methods section "Emergence" (L323-325 in tracked changes). The text is now much clearer and I agree that what they have done makes physical sense. I think, though, that

writing the equation as $\sigma_{ens}(N, \text{avg}2080-2099)$, i.e. with the average 2080-2099 inside the brackets suggests that the 20-year average is performed before taking the standard deviation, i.e. that they take the standard deviation of 20-year means. I don't think this is what they have done, so I would suggest carefully checking the notation of the equation. I think it should be something like $\text{avg}(\sigma_{ens}, 2080..2099)(N, t)$

Thanks for this important hint concerning the notation of the equation on emergence. We fully agree and have changed the notation as suggested.

Reviewer #3 (Remarks to the Author):

We thank Chris Folland for his continuous support to improve our manuscript and for this positive evaluation.

General

The authors have made a very good responses to the reviews so I am happy, subject to a small editorial comment below, to see this paper published as it is. The point not accepted for action about the over-sensitivity of the models to volcanic eruptions in the past would I think make modelled local temperature variability somewhat too great qualitatively. But I agree this cannot be quantified from existing data and this problem does not affect the key results of the paper.

This area of variability has recently experienced interesting revisions, as it became clear that if the EL Nino sequence in observed volcanic eruptions is reproduced in a model e.g. by sub-selection from a large ensemble or nudging, the difference is now much smaller in observed and modelled amplitude (see Lehner et al., 2016).

Reference: Lehner, F., Schurer A. P., Hegerl G. C., Deser C., Froehlicher T. L. (2016): Importance of ENSO phase during volcanic eruptions for detection and attribution. Geophys. Res. Lett., Geophys. Res. Lett. 43, 2851-2858. doi: 10.1002/2016GL067935.

Small Editorial point

In Fig. 4b, right of top diagram, what does the C mean in πC ? I expected π .

We referred to the range of variability estimated from the preindustrial control simulation (πC) of each model. As suggested, we now changed the terms in Figure 4b to “above/below preindustrial range of internal variability”.

Chris Folland Aug 4 2021